

# WHAT-IF: an open-source decision support tool for water infrastructure investment planning within the Water-Energy-Food-Climate Nexus

Raphaël Payet-Burin[1,2], Mikkel Kromann[2], Silvio Pereira-Cardenal[2], Kenneth Strzepek[3], Peter Bauer-Gottwein[1]

[1]Department of Environmental Engineering, Technical University of Denmark, Kgs. Lyngby, 2800, Denmark
[2]COWI A/S, Kgs. Lyngby, 2800, Denmark
[3]MIT, Joint Program Sci & Policy Global Change, 77 Massachusetts Ave, Cambridge, MA 02139, USA

*Correspondence to*: Raphaël Payet-Burin (rapp@env.dtu.dk)

**Abstract.** Water infrastructure investment planning must consider the interdependencies within the water-energy-food nexus. Moreover, uncertain future climate, evolving socio-economic context, and stakeholders with conflicting interests, lead to a highly complex decision problem. Therefore, there is a need for decision support tools to objectively determine the value of investments, considering the impacts on different groups of actors, and the risks linked to uncertainties. We present a new open-source hydroeconomic optimization model, linking in a holistic framework, representations of the water, agriculture, and power systems. The model represents the joint development of nexus-related infrastructure and policies and evaluates their economic impact, as well as the risks linked to uncertainties in future climate and socio-economic development. We apply the methodology in the Zambezi River Basin, a major African basin shared by eight countries, in which multiple investment opportunities exist, including new hydropower plants, new or resized reservoirs, development of irrigation agriculture, and investments into the power grid. We show that the linkage of the different systems is crucial to evaluate impacts of climate change and socio-economic development, which will ultimately influence investment decisions. We find that climate change could induce economic losses up to 2.3 billion dollars per year on the current system. We show that the value of the hydropower development plan is sensitive to future fuel prices, carbon pricing policies, the capital cost of solar technologies, and climate change. Similarly, we show that the value of the irrigation development plan is sensitive to the evolution of crop yields, world market crop prices and climate change. Finally, we evaluate the opportunity costs of restoring the natural floods in the Zambezi delta; we find limited economic trade-offs under the current climate, but potentially major trade-offs with irrigation and hydropower generation under climate change.

## 1 Introduction

Having established Integrated Water Resources Management Plans, many countries and river basins around the world are now planning to formulate water infrastructure development plans. These plans will help countries and regions realize the potential of their water resources – including agriculture, energy generation, and tourism – while preserving the environment.



Infrastructure investments will contribute to multiple Sustainable Development Goals (United Nations, 2015), such as : End Poverty (1), Zero Hunger (2), Clean and affordable energy for all (6), Clean and available water for all (7), Sustainable economic growth (8), and Climate Action (13). However, formulating these investment plans is a complex process involving competing objectives, upstream-downstream trade-offs, interactions between investments, multiple stakeholders and uncertainty related to socio-economic changes and future climate. In particular, it requires evaluating the interactions in the Water-Energy-Food (WEF) nexus.

The WEF nexus is an expanding topic in the literature. Albrecht et al. (2018) provide a systematic review of nexus approaches; Bazilian et al. (2011), McCarl et al. (2017) and Miralles-Wilhelm (2016) consider modelling and research challenges and Khan et al. (2017) focus on the water and energy sectors. Nexus studies cover resource use efficiency, institutional analysis, decision-making, and policy integration, using a broad range of methods such as integrated models, input-output analysis, Life Cycle Assessment and stakeholder engagement. In general, they aim to identify trade-offs between the different sectors and to support the development of cross-sectorial solutions, which produce additional benefits in comparison to single resource assessments (Albrecht et al., 2018). There are two strategies to model the interdependencies in the nexus: one is to couple well-established single system models where the output of the one feeds the input of the other in a one-way or iterative process (e.g. Howells et al. (2013) and Kraucunas et al. (2015)); another is the holistic approach which internally represents all interactions within a single model (e.g. Kahil et al. (2018) and Khan et al. (2018)). The advantage of coupling models is that it simplifies communication among stakeholders in different areas that can use their respective tools and enables a more detailed representation of single systems, while the holistic approach better represents interrelations and is more effective in an optimization framework. A challenge in both cases is to represent the diversity of the scales (spatial, temporal and political) where interactions occur (McCarl et al., 2017). While there is no approach that can fit all purposes, few models consider a spatial and temporal scale that can represent the interactions of water infrastructure with the WEF nexus.

Hydroeconomic optimization models (HOM) have developed into potential decision support tools for basin-scale water resources management over the past decade (see reviews by Bauer-Gottwein et al. (2017) and Harou et al. (2009)). They have been used to analyse water infrastructure investments, reservoir release scheduling and transboundary resources sharing problems. (e.g. Dogan et al. (2018), Draper et al. (2003), Goor et al. (2010), and Tilmant and Kinzelbach (2012)). Models include a representation of the regional-scale flow network; water availability, water uses and willingness-to-pay. By associating an economic impact to each decision, the complex multi-objective management problem becomes a simpler single-objective problem. Traditionally, agricultural and energy water users are represented with an exogenous demand and willingness-to-pay for water. Therefore, classic hydroeconomic models are able to analyse trade-offs and synergies between water users, but are not as effective in terms of representing dynamic interactions between infrastructure, policies, and commodity markets. For example, increased production of a commodity may lead to a lower market price of the commodity and thus to a lower willingness-to-pay for water. On the other hand, nexus models, particularly energy centred models (e.g. OSeMOSYS (Howells et al., 2011) and TIAM-FR (Dubreuil et al., 2013)) tend to ignore the spatial and temporal scale of water availability and therefore may overlook water scarcity problems (Khan et al., 2017).



Over the past 20 years, an increasing amount of legal and policy frameworks for transboundary water management have been implemented in internationally shared water courses (Qwist-Hoffmannn and McIntyre, 2016). River basin organisations are intended to facilitate the application of such mechanisms. In the Southern African Development Community (SADC), a state willing to implement a project, needs to notify potentially affected riparian states, including a description of the projects and

its potential impacts (SADC, 2000). Furthermore, most international financial institutions (e.g. AfDB, World Bank) require "No-objection" from riparian states to fund projects. Therefore, there is a need for decision support tools to objectively determine the impacts of WEF related projects on transboundary watersheds.

In this study, we developed a new open-source decision support tool for water infrastructure investment planning, based on a hydroeconomic optimization model in a nexus framework. The tool can represent political boundaries, the joint development

of WEF infrastructure and policies, and uncertainty in future climate and socio-technical changes. It aims to provide quantitative answers to the following prototypical questions:

-What is the economic impact of a given project or set of projects? Which is the best alternative among different investment plans?

-What are the synergies or trade-offs between investments and/or policies in different sectors? (e. g. what are the trade-offs

between hydropower, irrigation development plans and ecosystem preservation)

-What are the risks linked to uncertainty in future climate and socio-economic changes? Which investments and policies will be more robust to a range of future conditions?

This article is structured as follow: firstly, Sect. 2 presents the general modelling framework and details the representations of the water, energy, and food systems and the economic optimization. Secondly, we illustrate an application of the model on the

Zambezi river basin, where water resources of the eight riparian countries play a central role in the regional economy and are critical to sustainable economic growth and poverty reduction. Section 3 shows the input dataset for the study case, as well as the investment opportunities such as new hydropower plants, new or resized reservoirs, development of irrigation agriculture, and investments into the power grid. We show in Sect. 4 how the model answers the previous questions to assist decision-making. Finally, we discuss the limitations and improvement opportunities of the modelling approach in Sect. 5.

**2 Methodology of the decision support tool**

Figure 1 provides an overview of the decision support tool methodology, with the representation of the WEF subsystems and their main components. Subsystem representations are based on the concepts used in models such as WEAP (Yates et al., 2005) for the hydrology and water management, OSeMOSYS (Howells et al., 2011) for energy systems and IMPACT (Robinson et al., 2015) for agriculture. Subsystems are presented as blocks only for explanation purposes; the model internally

represents the interrelations in the nexus. The core component is the economic optimization framework, using a single objective function taking into account the different production costs, transaction costs and supply benefits of the different WEF commodities. In welfare economic terms, the objective function maximizes the sum of the total consumer and producer



surpluses, where the consumer surplus is the difference between the consumers' willingness to pay and the market price, and the producer surplus is the difference between the market price and the producers' production costs (Krugman and Wells, 2005). In contrast to simulation models that are rule-based (such as WEAP), the model finds the optimal water, agriculture and energy management decisions, considering trade-offs and synergies between them. The optimization framework simulates

adaptation to new infrastructure and policies, climate change, and socio-economic development. Conversely, in a rule-based simulation framework, allocation rules are usually based on the current socio-economic conditions or new rules are estimated, which may lead to suboptimal allocation decisions and underestimation of project benefits (Pereira-Cardenal et al., 2016). The optimization approach is based on a perfect foresight formulation, assuming that optimal decisions are found with full knowledge of the planning period; limitations of this common approach in sectoral planning models are discussed in Sect. 5.

The main outputs are economic indicators (such as market prices, consumer and producer surpluses), as well as water, energy and agriculture management decisions (such as supply, consumption, storage, production and transport). To calculate the economic impacts of an investment plan or a specific project, with/without analyses are performed, and different options can be compared. With/without analyses tend to overestimate benefits when no alternative is represented, particularly in growing economies (Griffin, 2008). Therefore, the model also integrates capacity expansion representations for the energy system and

alternative supply sources for agriculture, such as import or rainfed agriculture. To represent uncertainties linked to future climate or socio-economic development, the same investment plan or infrastructure is evaluated for different scenarios defined by the user. Hence, the decision support tool can be used as a discussion platform for stakeholders, answering questions such as "What are the economic impacts on producers and consumers of crops, energy and water of the projects?", "What if in the future available water resources are reduced because of climate change?" or "How robust is a plan considering uncertainties

in socio-economic development?".





**Figure 1: Conceptual framework of the decision support tool.** The water, agriculture and energy system are connected in the economic optimization framework. The blocks represent the different processes used in the model to represent the water, energy and food systems, while the circle contains the economic and physical interactions. The block representation is only for explanatory purposes; interactions are solved in a holistic approach**.**

The model is open-source and coded in the python programming language, using the pyomo modelling framework (Hart et al., 2017). The code and installation instructions can be found on Githublink. The model can be connected to different open-source or commercial solvers and input data is organized in MS Excel spreadsheets. We adopt a general framework that is study case





independent. Depending on the context, the availability of data, and the questions that the decision support tool is supposed to answer, some elements can be relevant or not. For this reason, the model is holistic in its resolution, but modular in its formulation, the user can activate or deactivate different modules and new modules representing relevant interrelations are easy to add.

In the following sections, we describe the equations and parameters used in the internal modules represented in Figure 1. For the practical implementation of the modules and their parametrization, the reader is referred to Sect. 3 for the Zambezi study case.

## 2.1 Water management

The water module represents hydrology and water management. The basic hydrological time scale is at monthly time steps,
but this is not a fixed requirement. The river network is described by a node-based approach, where the modelled area is divided into catchments with corresponding precipitation, evapotranspiration, surface runoff and groundwater recharge. Water transfer channels form additional links to the river network. The water is stored and released from reservoirs and is allocated to water users, while lakes and groundwater are represented as linear reservoirs. Evaporative losses take place in the river network, reservoirs and lakes. Water supply costs and losses are also considered. Water users can be defined with a water
demand and an associated marginal value; however, agriculture users and hydropower have a dynamic demand and marginal value detailed in the agriculture and energy modules.

The water resource can have an important value for activities that are not directly represented in the model, such as ecosystems, tourism, fishing, and transportation. Rather than giving it an economic value that may be hard to define and very uncertain (Loucks et al., 2005), the environmental flows module enables to define minimum flow requirements that have to be guaranteed
in the river. For methods to quantify environmental flow requirements, see Tharme (2003).

Figure 2 shows the conceptual scheme of the water module, while Table 1 lists used indices, parameters and decision variables. In the following equations, indices are only detailed when they enhance comprehension and capital letters denote decision variables, while parameters are noted as lower case letters:

Water balance, for time step t, catchment c:

$$q_{\text{runoff}} + Q_{\text{baseflow}} + Q_{\text{in}} = V_{res}[t] - V_{res}[t-1] + E_{\text{W}} + \sum_{\text{users}} S_{\text{W}} \cdot \left( \frac{1}{1 - l_{\text{user}}} - r_{\text{user}} \right) + T_{\text{W}} + Q_{\text{out}} \qquad (1)$$

$$\sum_{\text{users}} S_{\text{W}} \cdot \frac{1}{1 - l_{user}} \leq Q_{\text{in}} + Q_{\text{runoff}} + Q_{\text{baseflow}} \qquad (2)$$

Where:

$$Q_{\text{in}} = \sum_{\substack{\text{upstream} \\ \text{catchments}}} Q_{\text{out}} \cdot (1 - l_{\text{river}}) + \sum_{\substack{\text{incoming} \\ \text{transfers}}} T_{\text{W}} \cdot (1 - l_{\text{W,trans}}) \qquad (3)$$

$$E_{\text{W}} = (e_{\text{T0}} - p) \cdot (k_{\text{W}} \cdot \frac{V_{\text{W}}[t] + V_{\text{W}}[t-1]}{2} + a_{\text{W}}) \qquad (4)$$





$$Q_{\text{baseflow}} = V_{\text{GW}}[t-1] \cdot (1 - e^{-\alpha_{\text{GW}}}) + \left(q_{\text{recharge}} - S_{\text{GW}}\right) \cdot (1 - \frac{1 - e^{-\alpha_{\text{GW}}}}{\alpha_{\text{GW}}}) \qquad (5)$$

The water balance at the catchment boundaries Eq. (1) equals local runoff, groundwater base flow, and upstream inflows with reservoir storage variation, reservoir evaporation, water supply to catchment users, water transfer, and river outflow. Equation (2) ensures that the releases of the downstream reservoir are not allocated to upstream demand and assumes that return flows are not available for users inside the catchment. The catchment upstream inflow Eq. (3) is defined as the sum of outflows from upstream catchments considering losses in the river, and incoming transfer flows. The evaporative losses in the reservoirs Eq. (4), are based on a linear relation between the reservoir area and volume (parametrized by $k^{\text{res}}$ and $A^{\text{res}}$), using the average volume in a time period.

Linear reservoirs:

$$V_{\text{W}}[t] = V_{\text{W}}[t-1] + Q_{\text{in}} - E_{\text{W}} - \alpha_{\text{W}} \cdot \frac{V_{\text{W}}[t] + V_{\text{W}}[t-1]}{2} \qquad (6)$$

$$V_{\text{GW}}[t] = V_{\text{GW}}[t-1] \cdot e^{-\alpha_{\text{GW}}} + \left(q_{\text{recharge}} - S_{\text{GW}}\right) \cdot \frac{1 - e^{-\alpha_{\text{GW}}}}{\alpha_{\text{GW}}} \qquad (7)$$

Equation (6) only applies to linear reservoirs such as lakes, for which outflow is proportional to the storage volume. It assumes that a separate catchment is defined for the lake. The groundwater volume equation Eq. (7), is the analytical solution of the differential equation $\partial V_{GW} / \partial t = Q_{\text{recharge}} - S_{\text{GW}} - \alpha_{\text{GW}} \cdot V_{\text{GW}}$ where $Q_{\text{recharge}}$ and $S_{\text{GW}}$ are assumed to be constant during a time step. A similar expression could be used for linear reservoirs in Eq. (6), however, in this case the reservoir evaporation in the water balance would need to be differentiated between controlled and linear reservoirs, therefore we use the discrete solution for all reservoirs.

Capacity constraints:

$$S_{\text{W}} \le d_{\text{W}} \qquad (8)$$

$$V_{\text{W}} \le \bar{V}_{\text{W}} \qquad (9)$$

$$T_{\text{W}} \le \bar{T}_{\text{W,trans}} \qquad (10)$$

Equations (8), (9) and (10) represent the maximum demand of water users and the capacity limit of the reservoirs and transfer schemes.

Water supply costs and benefits:

$$\text{WSC} = \sum_{t,u} c_{\text{W}} \cdot S_{\text{W}}[t,u] + c_{\text{GW}} \cdot S_{\text{GW}}[t,u] \qquad (11)$$

$$\text{WSB} = \sum_{t,u} b_{\text{W}} \cdot (S_{\text{W}}[t,u] + S_{\text{GW}}[t,u]) \qquad (12)$$

Water supply costs in Eq. (11) represent the costs of supplying water to the users (e.g. pumping costs), they differ for surface water and groundwater. The water supply benefits in Eq. (12) represent the value of water allocations for non-agricultural users as the value of water for agriculture is endogenously determined in the agriculture production and crop market modules. The water supply costs and benefits are accounted for in the objective function of the model (Sect. 2.6).



Environmental flow requirements:

$$Q_{\text{out}} \geq q_{\text{env}} \qquad (13)$$

Equation (13) represents the minimum flow at the catchment outlet to preserve the ecosystems or other related activities. As the available runoff may go below the requirement, the constraint can be adapted to available runoff. Some environmental policies are designed to be respected only most of the time (e.g. 4 out of 5 years), such requirements can also be defined in the
5   model.

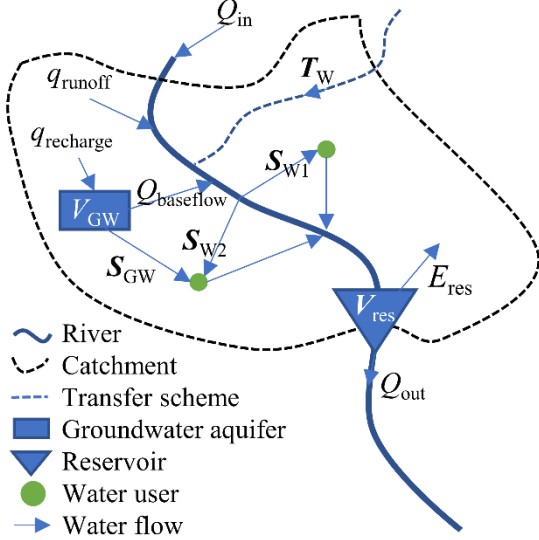

**Figure 2: Conceptual scheme of the water management module.** The scheme shows the main parameters and decision variables for a catchment with a groundwater aquifer, a reservoir, one incoming transfer scheme and two water users.

**Table 1: Water management indices, parameters and decision variables.** Bold characters denote independent decision variables.

| Notation | Description | dim | unit |
|---|---|---|---|
| **Indices** | | | |
| t | Time steps | | month |
| c | Catchments | | |
| aq | Groundwater aquifers | | |
| ts | Transfer schemes | | |
| r | Reservoirs | | |
| u | Water users | | |
| **Parameters** | | | |
| $q_{runoff}$ | Runoff | t, c | m³ month⁻¹ |
| $q_{recharge}$ | Groundwater recharge | t, aq | m³ month⁻¹ |



| | | | |
|---|---|---|---|
| $p$ | Precipitation | t, c | m³ month⁻¹ ha⁻¹ |
| $e_{T0}$ | Reference evapotranspiration | t, c | m³ month⁻¹ ha⁻¹ |
| $l_{river}$ | Water losses in the river | c | - |
| $\bar{V}_W$ | Reservoir storage capacity | r | m³ |
| $k_W$ | Volume-Area linear coefficient | r | ha m⁻³ |
| $a_W$ | Volume-Area linear constant | r | ha |
| $\alpha_W$ | Reservoir outflow coefficient | r | month⁻¹ |
| $\alpha_{GW}$ | Groundwater outflow coefficient | aq | month⁻¹ |
| $l_{W,trans}$ | Transfer scheme loss rate | ts | - |
| $\bar{T}_{W,trans}$ | Capacity of the transfer scheme | ts | m³ month⁻¹ |
| $q_{env}$ | Environmental flow requirement | t, c | m³ month⁻¹ |
| $d_W$ | User net water demand | t, u | m³ month⁻¹ |
| $l_{user}$ | User supply loss rate | u | - |
| $r_{user}$ | User return flow rate | u | - |
| $b_W$ | Marginal value of water use | u | $ m⁻³ |
| $c_W$ | Cost of surface water supply | u | $ m⁻³ |
| $c_{GW}$ | Cost of groundwater water supply | u | $ m⁻³ |
| **Decision variables** | | | |
| $Q_{in,out}$ | Inlet and Outlet flow, fixed in Eq. (1) | t, c | m³ month⁻¹ |
| $S_W$ | Surface water supply | t, u | m³ month⁻¹ |
| $S_{GW}$ | Groundwater supply | t, u | m³ month⁻¹ |
| $V_{res}$ | Reservoir storage volume | t, r | m³ |
| $T_W$ | Transfer flow | t, ts | m³ month⁻¹ |

## 2.2 Agriculture production

The agriculture module computes local water demand for agriculture and production of crops depending on water allocation and rainfall. Farming zones represent agriculture areas with a specific farm type, have a limited area and belong to a catchment and a country. Farm types can represent different soil qualities, fertilizer/pesticides inputs and availability of irrigation and

5    drainage systems. Farm types define the potential yields, cultivation and infrastructure costs, they can be used to represent different kinds of agriculture, such as rainfed, irrigated and subsistence agriculture or differences among the countries/regions depending on available data and the user's interest. Crops (as a traded commodity) are produced at the yearly time step by cultures. Cultures are divided into growth phases (e.g. initial, crop-development, mid-season and late season) which take place during a specific period of the year. Water requirements by cultures are estimated using the FAO 56 method (Allen et al.,

10    1998), with the reference evapotranspiration and a culture and phase specific crop coefficient. The relation between water





allocated to cultures and yield is estimated using the additive yield water response function based on the FAO 33 method (Doorenbos and Kassam, 1979). In a farming zone the same area can be used by several cultures during different periods of the year, representing multiple harvests per year; the schedules are defined by the user. The model either finds the optimal crop choice per year or assumes a fixed crop distribution for the entire simulation period. However, additional constraints such as

maximum area per culture and farming zone can be used to represent physical, institutional or economic constraints which are otherwise not included in the modelling framework. Crop production costs represent costs of infrastructure, machinery, labour, land, chemicals and fertilizers, depending on the culture and farm type. Table 2 shows the indices, parameters and decision variables used in the following equations:

Land use, for year y, farming zone fz:

$$\sum_{\text{cultures}} A \le \bar{A} \tag{14}$$

Equation (14) represents the land use constraint per farming zone, cultures on the same area at different period of the year are counted once.

Linearized additive yield water response function, for year y, farming zone fz, culture cul:

$$P_{\text{C}} = y \cdot \sum_{\text{pt}} \left( A[\text{pt}] \cdot (1 - \sum_{\text{ps}} k_{\text{Y}}[\text{ps}] \cdot (1 - m[\text{pt}, \text{ps}])) \right) \tag{15}$$

The yield water response function (Doorenbos and Kassam, 1979) expresses that crop production is proportional to the maximum yield ($y$), corrected by the yield response factor ($k_{\text{Y}}$), which characterizes how the yield responds to water stress in

the different growth phases. This expression is not linear as it is the product of two decision variables (cultivated area and water supply to cultures). The equation is linearized in Eq. (15) by linking the crop water demand satisfaction and the cultivated area in a single decision variable $A[\text{pt}]$ where pt represents the different demand satisfaction paths and $m[\text{pt}, \text{ps}]$ the associated demand satisfaction rates for the different growth phases. Appendix A details the derivation of Eq. (15) and Eq. (16).

Water supply, for year y, farming zone fz, time step t:

$$S_{\text{W}}[\text{t}, \text{fz}] + S_{\text{GW}}[\text{t}, \text{fz}] = \frac{1}{1 - r_{\text{user}}[fz]} \cdot \sum_{\text{cul}, \text{ps}, \text{pt}} a[\text{cul}, \text{ps}, \text{t}] \cdot A[\text{cul}, \text{pt}] \cdot \max(0, k_{\text{c}}[\text{cul}, \text{ps}] \cdot e_{\text{T0}}[\text{t}] \cdot m[\text{pt}, \text{ps}] - p[\text{t}]) \tag{16}$$

The water supply Eq. (16), is the link between the water and agriculture module. The farming zones are considered as water users and their surface and groundwater water supply ($S_{\text{W}} + S_{\text{GW}}$) is determined by the cultivated area ($A$), the water demand by cultures based on FAO 56 ($k_{\text{c}} \cdot e_{\text{T0}}$), the chosen demand satisfaction path of the cultures ($m$), and the precipitation ($p$). The factor $a$ represents the share of the time step falling into a specific growth phase for the different cultures and $r_{\text{user}}$ is the leaching factor of the farming zone to avoid salinization of the soil.

Crop production costs:

$$\text{CPC} = \sum_{\text{y}, \text{fz}, \text{cul}} c_{\text{cult}} \cdot A \tag{17}$$





Crop production costs are assumed to be proportional to the cultivated area and are accounted for in the objective function of the model (Sect. 2.6).

**Table 2: Agriculture Production indices, parameters and decision variables**

| Notation | Description | dim | unit |
|---|---|---|---|
| **Indices** | | | |
| y | Years | | |
| fz | Farming zones | | |
| ft | Farm types | | |
| cr | Crops | | |
| cul | Cultures | | |
| ps | Growth phases | | |
| pt | Demand satisfaction paths | | |
| **Parameters** | | | |
| $\bar{A}$ | Land capacity | fz | ha |
| $y$ | Potential yield | ft, cul | t ha$^{-1}$ |
| $a$ | Month to phase coefficient | t,ps,cul | - |
| $k_c$ | Single crop coefficient | ps, cul | - |
| $k_Y$ | Yield water response factor | ps, cul | - |
| $c_{cult}$ | Cultivation costs | ft, cul | \$ ha$^{-1}$ |
| **Decision variables** | | | |
| $A$ | Cultivated area | y, fz, cul, pt | ha |
| $P_C$ | Crop production, fixed in Eq. (15) | y, fz, cul | t yr$^{-1}$ |

**2.3 Crop markets**

5   The crop market module represents the local demand, transport, and trade of crops. Crop markets are characterized by a demand, a marginal value and a demand elasticity for the different crops. A minimum supply requirement can be defined, to represent food security constraints. Crops produced in the farming zones are transported between crop markets through transport routes, with associated costs and losses. External markets can be introduced to represent imports and exports out of the study area. These markets behave as the other crop markets, but their crop production is represented through an external

10   crop production function which does not depend on farming zones (the function is assumed to be infinite and perfectly inelastic). Table 3 shows the indices, parameters and decision variables used in following equations:

Crop balance, for year y, crop market cm, crop cr:





$$\sum_{\substack{\text{local} \\ \text{farming zones}}} P_C + P_{C,ext} + \sum_{\text{imports}} T_C \cdot (1 - l_{C,trans}) = S_C + \sum_{\text{exports}} T_C \qquad (18)$$

In Eq. (18) the crop production ($P_C$) of local farming zones and external production ($P_{C,ext}$) (for markets out of the study area) plus crop imports ($T_C$) from other markets equals the crop supply to the local market demand ($S_C$) plus crop exports ($T_C$) towards other markets.

Crop demand and food security constraint, for year y, crop market cm, crop cr:

$$S_C \le d_C \qquad (19)$$

$$S_C \ge d_{min} \qquad (20)$$

5    In Eq. (19) the crop supply ($S_C$) is limited to the demand ($d_C$) of the crop market. Equation (20) represents the minimum supply of crops ($d_{min}$) that must be fulfilled to ensure food security. The demand elasticity for crops is represented by a stepwise function, as described in Appendix B, therefore the demand and value are divided in demand steps (cds). The demand elasticity represents the fact that willingness to pay for crops is decreasing with increasing crop demand.

Crop supply benefits and crop supply costs:

$$CSB = \sum_{y,cm,cr,cds} b_C[cds] \cdot S_C[cds] \qquad (21)$$

$$CSC = \sum_{y,cm,cr} c_{ext} \cdot P_{C,ext} + \sum_{y,tr,cr} c_{C,trans} \cdot T_C \qquad (22)$$

10   The crop supply benefits Eq. (21) and costs Eq. (22) are used in the objective function of the model (Sect. 2.6). The benefits represent the value for consumers, the costs are the external production costs and the transaction costs among crop markets.

**Table 3: Crop markets indices, parameters and decision variables**

| Notation | Description | dim | unit |
|---|---|---|---|
| **Indices** | | | |
| cm | Crop markets | | |
| cds | Crop demand steps | | |
| tr | Transport routes | | |
| **Parameters** | | | |
| $d_C$ | Crop demand | cm, cds | t yr$^{-1}$ |
| $d_{min}$ | Crop minimum demand | cm | t yr$^{-1}$ |
| $l_{C,trans}$ | Crop transport loss rate | tr, cr | - |
| $b_C$ | Crop marginal value | cm, cr, cds | $ t$^{-1}$ |
| $c_{ext}$ | External supply costs | cm, cr | $ t$^{-1}$ |
| $c_{C,trans}$ | Crop transaction costs | tr, cr | $ t$^{-1}$ |
| **Decision variables** | | | |
| $S_C$ | Crop supply | y, cm, cr | t yr$^{-1}$ |





| $T_C$ | Crop transport | y, tr, cr | t yr⁻¹ |
| $P_{C,ext}$ | Crop external production | y, cm, cr | t yr⁻¹ |

## 2.4 Energy production

The energy modules focus on electric energy, also called the "power system", and do not consider fuels for transportation, cooking or heating. Power is produced by hydropower turbines and other power plants (such as thermal, solar, wind and biomass). Hydropower turbines are either linked to a reservoir or are run-off-the-river and have associated operation costs and
water-energy equivalent factors. Other power plants are defined by their efficiency, fuel use, operation costs and production capacity. In addition, generic power technologies represent additional capacity that can be invested in, similarly to capacity expansion models (e.g. Howells et al. (2011)). They have associated capacity construction costs, fixed and variable operational costs, fuel use and efficiencies that can be defined for every power market (see Sect. 2.5 for power markets). "Other power plants" and "generic power technologies" are represented in a similar way; the main difference is that the first can be used to
represent specific existing or planned power production units, while the second represents potential technologies available to the capacity expansion model. Fuels represent the different natural resources that can be used to produce energy (e.g. coal, gas or sun); fuel consumption is determined by the power plant's efficiency and a fuel price can be defined per power market. $CO_2$ emissions are associated to different fuels, which lead to $CO_2$ emission costs if a carbon cost is defined. Table 4 shows the indices, parameters and decision variables in the following equations:
Hydropower discharge and production, for time step t:

$$\sum_{\text{hydro turbines}} Q_{\text{hydro}} \leq Q_{\text{out}} \qquad (23)$$

$$P_{\text{hydro}} = \gamma \cdot Q_{\text{hydro}} \qquad (24)$$

In Eq. (23) the sum of the discharges through the hydropower turbines belonging to the same reservoir is lower or equal to the outflow of the reservoir $Q_{\text{out}}$, the difference being the spill of the reservoir. The same relation applies to run-off-the-river hydropower, except that the hydropower is not linked to a specific reservoir but to a catchment. The power production of hydropower turbines Eq. (24) assumes fixed head of the corresponding reservoir, where γ (kWh m⁻³) is the average water-
energy equivalent. The fixed head assumption leads to overestimated discharge capacity and hydropower production during droughts when reservoirs are at low levels. This assumption permits to keep the model linear; it can be relaxed by introducing mixed integer programming or non-linear constraints but comes at the cost of increased computational requirements.
Energy production costs:

$$EPC = OC + FC + CC \qquad (25)$$

$$OC = \sum_{t,ls,hp} c_{\text{om,hydro}} \cdot P_{\text{hydro}} + \sum_{t,ls,op} c_{\text{om,plant}} \cdot P_{\text{plant}} + \sum_{t,ls,pt} c_{\text{om,tech}} \cdot P_{\text{tech}} \qquad (26)$$

$$FC = \sum_{fu} (c_{\text{fuel}} + c_{CO2} \cdot e_{CO2}) \cdot \left( \sum_{op \in fu} P_{\text{plant}} \big/ e_{\text{plant}} + \sum_{pt \in fu} P_{\text{tech}} \big/ e_{\text{tech}} \right) \qquad (27)$$





$$CC = \sum_y C_{\text{tech}} \cdot \left( {^{C_{\text{cap,tech}}}} / {_{t_{\text{life}}}} + c_{\text{fix,tech}} \right) \tag{28}$$

The energy production costs (EPC) in Eq. (25) are the sum of the marginal operational costs (OC), the fuel consumption and $CO_2$ emission costs (FC) and the capacity expansion costs (CC), they are taken into account in the objective function of the model (Sect. 2.6).

**Table 4: Power production indices, parameters and decision variables**

| Notation | Description | dim | unit |
|---|---|---|---|
| **Indices** | | | |
| hp | Hydropower turbines | | |
| op | Other power plants | | |
| pt | Generic power technologies | | |
| fu | Fuels | | |
| **Parameters** | | | |
| $\gamma$ | Water-Energy equivalent | hp | kWh m$^{-3}$ |
| $e_{hydro}$ | Efficiency of hydropower plants | hp | - |
| $e_{plant}$ | Efficiency of other power plants | op | kWh kWh-fuel$^{-1}$ |
| $e_{tech}$ | Efficiency of power technologies | op | kWh kWh-fuel$^{-1}$ |
| $t_{life}$ | Lifetime of power technologies | pm, pt | yr |
| $e_{CO2}$ | $CO_2$ emission rate of fuels | fu | t-CO$_{2eq}$ kWh-fuel$^{-1}$ |
| $c_{om,hydro}$ | Operational costs of hydropower turbines | hp | \$ kWh$^{-1}$ |
| $c_{om,plant}$ | Operational costs of other power plants | op | \$ kWh$^{-1}$ |
| $c_{cap,tech}$ | Capital costs of generic technologies | pm, pt | \$ kW$^{-1}$ |
| $c_{fix,tech}$ | Fix operational costs of generic technologies | pm, pt | \$ kW$^{-1}$ yr$^{-1}$ |
| $c_{om,tech}$ | Variable operational costs of generic technologies | pm, pt | \$ kWh$^{-1}$ |
| $c_{fuel}$ | Fuel costs | pm, fu | \$ kWh-fuel$^{-1}$ |
| $c_{CO2}$ | $CO_2$ emission costs | - | \$ t-CO$_{2eq}^{-1}$ |
| **Decision variables** | | | |
| $Q_{hydro}$ | Discharge through hydropower turbines | t, ls, hp | m$^3$ month$^{-1}$ |
| $P_{hydro}$ | Hydropower production, fixed in Eq. (24) | t, ls, hp | kWh month$^{-1}$ |
| $P_{plant}$ | Other power plant energy production | t, ls, op | kWh month$^{-1}$ |
| $C_{tech}$ | Generic technology capacity expansion | t, pm, pt | kW |
| $P_{tech}$ | Generic technology production | t, ls, pm, pt | kWh month$^{-1}$ |



## 2.5 Energy markets

The power market module accounts for the power network and the power demand. Power markets define the resolution of the power network and the power demand, they can be defined nationally or regionally. As for crop markets, they are characterized by a demand and marginal value for power, however demand is assumed to be perfectly inelastic. Transmission lines carry

energy between power markets with associated costs and losses and a limited capacity. This corresponds to a "transport model" or "transhipment model", which does not consider reactive power flows and voltage angles, but is commonly used for planning energy systems as it requires less data and computation time than AC or DC power flow models (Krishnan et al., 2016). The base time scale for the power system is, as for the hydrology, the monthly time step. However, the power demand can be divided into different load segments (such as peak and base, day and night) defined by the user. Load segments are commonly

used in energy models with large time steps to better represent the effects of peaking demand (Palmintier, 2013). Some generic power technologies can have a limited capacity during specific load segments, this feature serves to represent renewable energies such as solar or wind (e.g. no solar energy is available at night, windy or less windy segments can be defined). Table 5 shows the indices, parameters and decision variables used in the following equations:

Energy balance, for time step t, load segment ls, power market pm:

$$\sum_{hp \in pm} P_{hydro} + \sum_{op \in pm} P_{plant} + \sum_{pt} P_{tech} + \sum_{tl \in imports} T_E \cdot (1 - l_{E,trans}) = S_E \frac{1}{1 - l_{E,supply}} + \sum_{tl \in exports} T_E \qquad (29)$$

Equation (29) is the energy balance at the power markets: the power produced by local hydropower, other power plants and additional capacity plus net imported power through the transmission network, equals the gross power supply to the local demand plus gross exported power.

Power demand, for time step t, load segment ls, power market pm:

$$S_E \leq d_E \cdot d_{load} \qquad (30)$$

In Eq. (30) the power supplied ($S_E$) is limited to the power demand of the corresponding load segment ($D_E \cdot d_{load}$).

Capacities, for time step t, and load segment ls:

$$P_{hydro} \leq \bar{P}_{hydro} \cdot t_{load} \qquad (31)$$

$$P_{plant} \leq \bar{P}_{plant} \cdot t_{load} \cdot e_{CF} \qquad (32)$$

$$P_{tech} \leq C_{tech} \cdot t_{load} \cdot e_{CF} \qquad (33)$$

$$T_E \leq \bar{T}_{E,trans} \cdot t_{load} \qquad (34)$$

In Eq. (31), (32) and (33) the hydropower, other power plants and generic technologies power productions (respectively $P_{hydro}$, $P_{plant}$ and $P_{tech}$) are limited by their capacities ($\bar{P}_{hydro}$, $\bar{P}_{plant}$ and $C_{tech}$) adjusted to the length of the load segment ($t_{load}$) and the eventual load segment capacity factor ($e_{CF}$), constraining some power technologies during the load segment. Similarly, the limited capacity of transmission lines is represented in Eq. (34).

Energy supply benefits and energy transmission costs:



$$\text{ESB} = \sum_{t,ls,pm} b_E \cdot S_E \qquad (35)$$

$$\text{ETC} = \sum_{t,ls,tl} c_{E,trans} \cdot T_E \qquad (36)$$

The energy supply benefits (ESB) Eq. (35) and transmission costs (ETC) Eq. (36) are used in the objective function of the model (Sect. 2.6).

**Table 5: Power market parameters and decision variables**

| Notation | Description | dim | unit |
|---|---|---|---|
| **Indices** | | | |
| pm | Power markets | | |
| ls | Load segments | | |
| **Parameters** | | | |
| $d_E$ | Power demand | t, pm | kWh month$^{-1}$ |
| $d_{load}$ | Share of the demand per load segment | ls | - |
| $t_{load}$ | Length of load segment | ls | h month$^{-1}$ |
| $e_{CF}$ | Load segment capacity factor | ls, pt | - |
| $\bar{P}_{hydro}$ | Capacity of hydropower turbine | hp | kW |
| $\bar{P}_{plant}$ | Capacity of other power plants | op | kW |
| $\bar{T}_{E,trans}$ | Capacity of the transmission line | tl | kW |
| $l_{E,trans}$ | Power transmission losses | tl | - |
| $l_{E,supply}$ | Local power supply losses | pm | - |
| $b_E$ | Marginal value of energy | pm | \$ kWh$^{-1}$ |
| $c_{E,trans}$ | Energy transmission costs | tl | \$ kWh$^{-1}$ |
| **Decision variables** | | | |
| $S_E$ | Net Power supply | t, ls, pm | kWh month$^{-1}$ |
| $T_E$ | Energy transmission | t, ls, tl | kWh month$^{-1}$ |

**2.6 Economic optimization**

5   The economic module is the objective function of the optimization model. The equations are solved to find the optimal water, agriculture and energy management decision variables minimizing the costs (/maximizing the benefits) resulting from previous modules while respecting the physical, political and economic constraints. In welfare economic terms, this corresponds to the maximization of the total consumer and producer surplus for all commodities represented: water, crops, and energy (see Krugman and Wells (2005) for details on consumer and producer surplus). According to the second welfare economic theorem,

10   any pareto optimal allocation can be reached by a competitive market. This means that the "centrally planned" solution from





the economic optimization module, is the same as the individual profit maximization solution, assuming that water, energy and crops could be traded on perfect markets.

The objective function φ to maximize is expressed as:

$$\varphi = WSB - WSC + CSB - CSC - CPC + ESB - ETC - EPC$$

5    Where WSB represents the water supply benefits Eq. (12), WSC the water supply costs Eq. (11), CSB the crop supply benefits Eq. (21), CSC the crop supply costs Eq. (22), CPC the crop production costs Eq. (17), ESB the energy supply benefits Eq. (35), ETC the energy transmission costs Eq. (36) and EPC the energy production costs which are the sum of the energy operational costs, fuel consumption and $CO_2$ emission costs and the capacity expansion costs Eq. (25).

The main outputs of the economic optimisation are the optimal decisions in terms of water, energy and agricultural 10    management and the resulting economic impacts on different groups of actors. Equally important outputs are the shadow prices of the constraints (also called duals) that reveal the equilibrium prices of the different commodities and give information about capacity constraints (e.g. the marginal value of additional storage or transmission capacity) that can help identify bottlenecks in the systems (Harou et al., 2009).

## 3 The Zambezi river basin study case

15    The Zambezi river plays a central role in the regional economy and is shared by eight riparian countries: Angola, Botswana, Malawi, Mozambique, Namibia, Tanzania, Zambia, and Zimbabwe. The countries formed the Zambezi River Commission (ZAMCOM) in 2014, which is the river basin organisation supporting transboundary water management. The water resource supports agriculture, fisheries, hydropower production, water supply and sanitation, navigation, tourism, industries and mining. The basin extends over almost 1.4 million square kilometres, sustaining the basic needs of 40 million people and a rich and 20    diverse natural environment. In the river basin, 77% of the population have access to safe and adequate water supply and 60% has access to adequate sanitation, which is above the Southern Africa averages (SADC et al., 2015). The area is mainly covered by forest and bush (75%), while cropland represents only 13% of the area, mainly rainfed, as less than 5% of the cropland is irrigated (SADC et al., 2015). The main source of energy is biomass, fulfilling 80% of the demand; a limited share of the population has access to grid electricity, ranging from 12% in Zambia to 40% in Zimbabwe (SADC et al., 2015). Population 25    is expected to grow rapidly, reaching 51 million in 2025 and 70 million in 2050, which will increase the demand for water, food and energy (SADC et al., 2015). Therefore, the water resources of the river basin are critical to sustainable economic growth and poverty reduction in the region.

The World Bank carried out the Multi-Sector Investment Opportunities Analysis (MSIOA) study (World Bank, 2010), that analyses the value of the hydropower and irrigation projects and trade-offs between them. The study finds that the hydropower 30    development plan is able to meet the region's current energy demand and that the implementation of the irrigation development plan would reduce the current hydropower production by 9%. Tilmant et al. (2012) also investigate trade-offs between hydropower and irrigation development in the Zambezi basin, using a stochastic hydroeconomic optimization formulation.



The study finds that some of the upstream irrigation projects are not viable if the downstream hydropower projects are developed. However, the study uses an exogenous price for hydropower and irrigation water, and, as the World Bank study, it does not consider climate or socio-economic changes. Beilfuss (2012) points out that most of the planned hydropower projects were evaluated using historical hydrologic data, not considering climate change and may therefore be economically not viable.

Furthermore, the study highlights the lack of consideration of the impact on ecosystems of large hydropower projects. In a further World Bank study, Cervigni et al. (2015) assess potential impacts of climate change on water infrastructure in sub-Saharan Africa. The study finds for the Zambezi that in the driest scenario hydropower production could decline by up to 60% and unmet irrigation demand increase by up to 25%. Focused on the power system, the IRENA (2013) study shows that 80% of capacity addition between 2010 and 2030 in the South African Power Pool (SAPP) could be renewable technologies. This

tendency is confirmed in Spalding-Fecher et al., (2017b) analysing electricity supply and demand scenarios for the SAPP power pool until 2070. Spalding-Fecher et al. (2017a) by combining the previous study with data from Cervigni et al. (2015) find that hydropower production could decline by 10-20% in a drying climate which could increase generation costs by 20 to 30% in the most hydropower-dependent countries. The agriculture system is, however, not part of the integrated analysis. In a broader perspective, the Zambezi Environment Outlook study (SADC et al., 2015), presents an integrated analysis of the

Zambezi river considering ecological, social and economic issues.

### 3.1 Hydrology, reservoirs and environmental flows

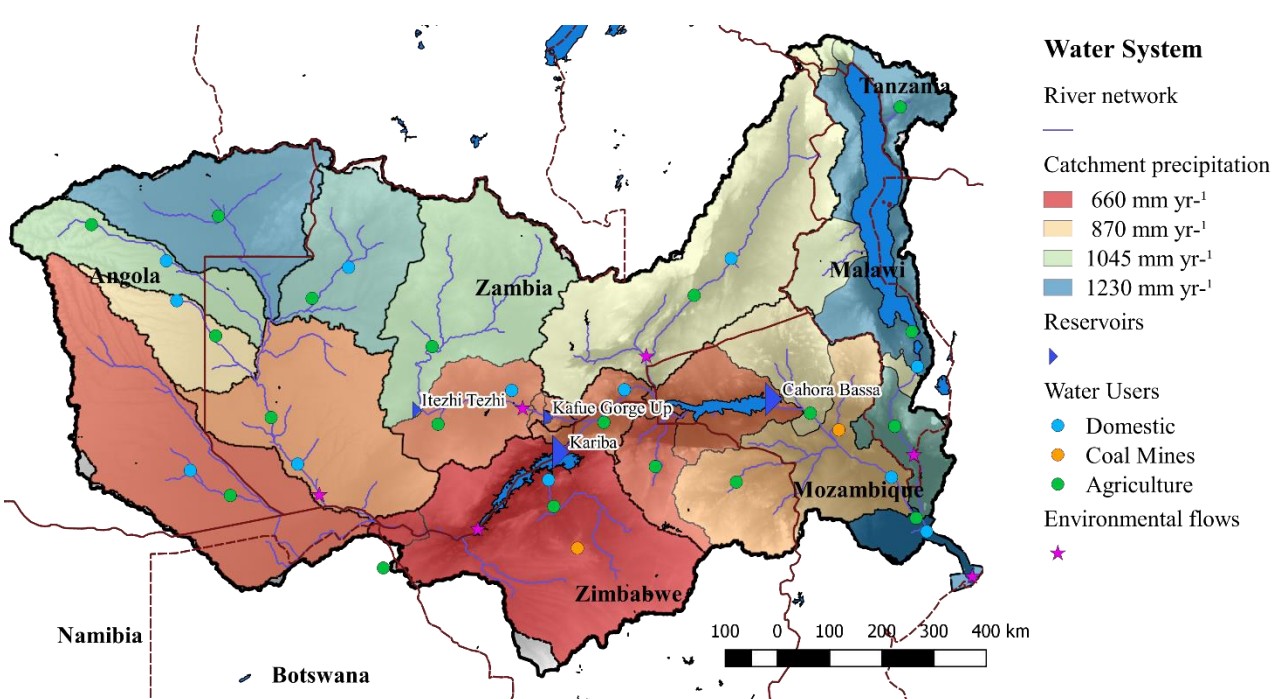

**Figure 3: The water system representation.** The river basin is divided into hydrological catchments defining the river network and a rainfall-runoff model gives water availability. Reservoirs can store and release water. Water users represent large non-agricultural

consumption, such as mining.





The hydrologic data used in this study is the same as the data used in Cervigni et al. (2015). The historical climate dataset is from Sheffield et al. (2006), and runoff is given by a lumped rainfall-runoff model from Strzepek et al. (2011). As the annual flow follows long term cycles, we use a 40 years time series, from 1960 to 1999: the years 1982-1998 are significantly below average and the years 1960-1982 are above average (Beilfuss, 2012). The rainfall-runoff model exogenously considers the

effect of wetlands that evaporate part of the river flow. Therefore, the impact of reservoir operations on wetland dynamics is not represented; however, only the Kafue flats are located downstream of a major reservoir (Itezhi-Tezhi) and upstream of other water users. According to World Bank (2010) groundwater is not overexploited in the river basin, furthermore there is almost no data available concerning groundwater, therefore, like in similar studies in the basin, groundwater is ignored in this study. The main reservoirs of the Zambezi river, Itezhi-Tezhi, Kariba and Cahora Bassa dam (Figure 3) have a total active

storage capacity of 127 000 million m³, slightly higher than the mean annual flow. Evaporation from the reservoirs is the main consumptive water use, ranging from 7 800 to 16 989 million m³ per year depending on the studies (Beilfuss, 2012; Euroconsult and Mott MacDonald, 2008; Tilmant et al., 2012), see Sect. 4.1 for more details. The volume-area relationships used to compute evaporation are derived from World Bank (2010). The main non-agricultural water users are the Gokwe and Moatize coal mines with 622 million m³ per year, other industrial and domestic water consumptions are relatively small and represent

only 175 million m³ per year (World Bank, 2010). Waters of the Zambezi sustain some fragile ecosystems, among them are: Kafue flats and Barotse Plain in Zambia, Mana Pools in Zambia and Zimbabwe, and Zambezi Delta in Mozambique. We do not represent the economic value of these, but use environmental flow requirements from (World Bank, 2010), which are based on two assumptions: flow should not drop below the low-flow level in dry years and average annual flow should not drop below 60% of the mean average annual flow. This constraint is applied at the Barotse floodplain, the Kafue Flats, the Luangwa

river, the lower shire, and the Zambezi delta (Figure 3).

### 3.2 Energy

The national power utilities of the Zambezi basin are members of the Southern African Power Pool (SAPP), which is the institution overseeing the power sector in southern Africa. The goal of the SAPP is to develop of a competitive electricity market in which power is traded in bilateral, forward physical, day ahead and intraday markets. The SAPP power pool is

dominated by South Africa which represents roughly 80 % of the demand and production (SAPP, 2015). Coal is the main source of power production (77%), followed by hydropower on the Zambezi and Congo river basins (21% of installed capacity), nuclear, gas and renewables representing only a minor share (SAPP, 2015). The members of the SAPP are interconnected with transmission lines, except for Angola, Malawi and Tanzania which are isolated. The demand for electricity is growing rapidly, and in recent years power shortfalls became common particularly in Mozambique, Malawi, South Africa,

Zambia and Zimbabwe (World Bank, 2010). Figure 4 shows an overview of the energy system representation.



### 3.2.1 Energy markets

To represent the energy system, we consider one power market per country (Figure 4), including South Africa which is the main power exchanger with the Zambezi basin. National power demands are found in SAPP (2015). We assume non-satisfied power demand is compensated by running fuel generators, so power curtailment costs are estimated at 240 $ MWh$^{-1}$. The

monthly energy demand is divided in two load segments: a base demand and a peak demand. Based on SAPP (2015), the peak load is during day and covers 70% of the total demand, while the base load is during night and covers 30% of the demand, both are assumed to cover half of the monthly time step. Energy demand is assumed to be perfectly inelastic, as most consumers do not have hour-by-hour metering, the price signal from the marginal cost of production is assumed to not reach the consumer. The transmission network is represented by aggregated transmission lines among countries that are connected, the transmission

capacity and loss rate are found in IRENA (2013), SAPP (2018) and World Bank (2010).

### 3.2.2 Energy production

We represent the existing hydropower plants and one aggregated power plant per country (Figure 4) representing the total non-hydropower generation capacity, using data from World Bank (2010). For hydropower plants, the water-energy equivalent factor is derived from turbine capacity in m³/s and power output in MW from World Bank (2010). In addition, three generic

technologies are available for additional investments: supercritical coal, combined cycle gas turbines (CCGT) and solar photovoltaic. Investment costs, fix and variable operation and maintenance costs, lifetime, and efficiency of these technologies are found in (IRENA, 2013), we assume the same parameters for all countries. However, gas and coal (fuels) costs vary among countries, depending on their local availability (IRENA, 2013). To represent intermittency constraints in a simplified way, solar energy is assumed to be unavailable during the base load segment (night), and the peak load segment (day) is divided in

two: days where solar energy can be produced and days where it can't. The length of these two load segments is adjusted to fit the load factor of 25 % used in (IRENA, 2013) for solar photovoltaic energy.





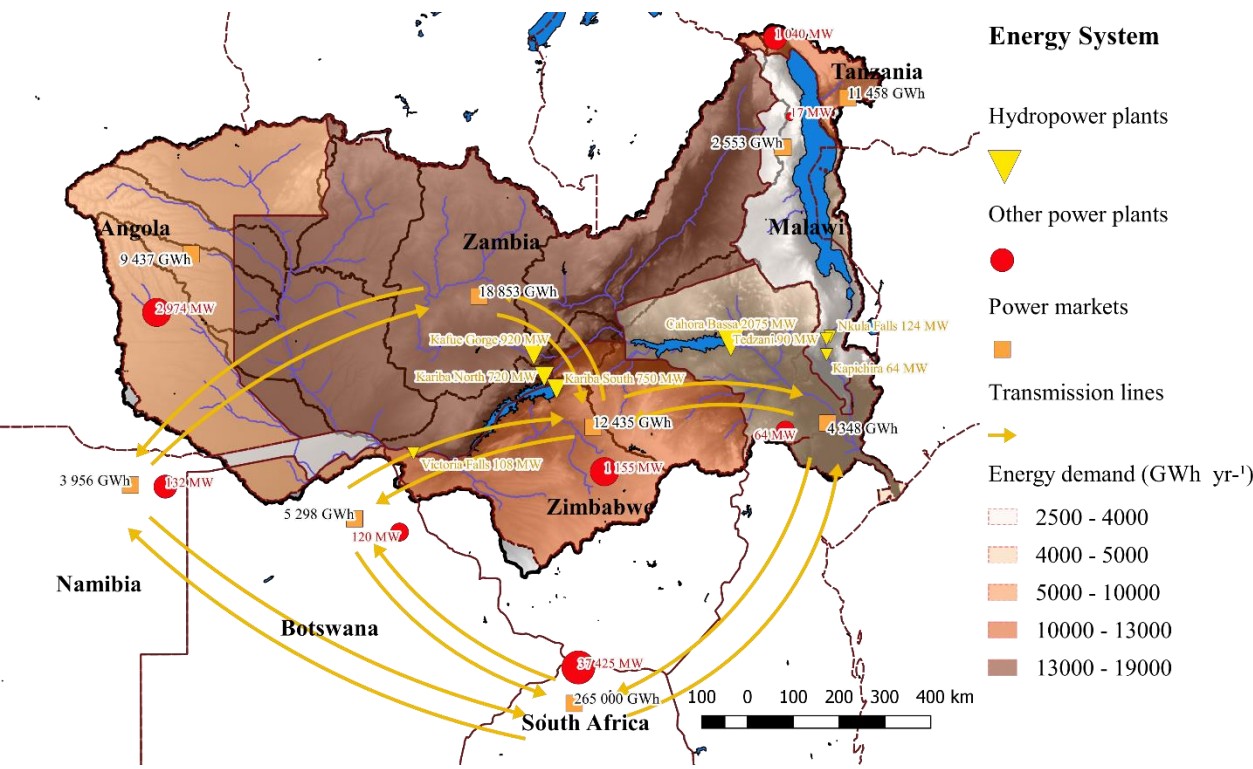

**Figure 4: The energy system representation.** Hydropower plants are represented individually while the remaining generating capacity is aggregated in a single power plant per country. Transmission lines among the countries permit power exchanges. Additional power generating capacity can be added in every power market by investing in one of the generic power technologies (Coal, Gas or Solar).

## 3.3 Agriculture

According to FAO (2018), the total production value of the agricultural sector in the Zambezi basin is around 6.7 billion dollars per year (the numbers are estimated by downscaling national statistics from 2010 to 2016 with the population ratio). Among these, 1.7 billion dollars is from exports and half of the exports are tobacco. The crop imports represent 1.2 billion dollars per year, wheat and rice being the most imported crops. Agricultural markets are heavily regulated by policies such as import or export bans and crop prices fixed by the governments, therefore little trade occurs among the Zambezi countries. The trade among Zambezi countries accounts for only 320 million dollars per year, and almost half of it is exports of maize and tobacco from Zambia to Zimbabwe.

### 3.3.1 Main crops and cultures

To represent the most significant crops in the agricultural module different aspects should be considered: the cultivated area per crop has the strongest impact on water demands for agriculture, however the value of agricultural production indicates which crops have the biggest economic impact. Another factor is which crops are mainly used for irrigated agriculture, as these will have a bigger impact on the nexus and irrigation projects. To simplify the model, some crops are grouped, which assumes





that crops in the same group are fully substitutable and have the same value. Table 6 shows all selected crops; cassava, maize and roots represent more than half of crop production, cultivated area and value of production. However, for irrigated agriculture the most important crops are cereals, rice, sugar cane and stimulants. Some of the crops can be produced by different cultures (e.g. summer and winter); the represented cultures are: cassava, potato (roots), wheat and sorghum (cereals), summer and winter maize, vegetables, sugarcane, summer and winter rice, fruits, groundnuts and soybeans (oilseeds), stimulants, summer and winter beans (pulses).

Potential yields of the different cultures are estimated as the maximum observed yield in each country from the FAO (2018) "Production quantity" and "Area harvested" data between the years 2000 and 2016. This assumes the maximum yield was obtained due to optimal hydrologic conditions, all other inputs being equal. In general, yields in Zambezi countries are lower than average expected yields because of very low inputs (World Bank, 2010). We consider four growing phases for all cultures (initial, crop-development, mid-season and late season). The corresponding crop coefficients (Kc) and yield water response coefficients (kY) used in the model to calculate the water requirements and the resulting yields are found in FAO 56 (Allen et al., 1998), World Bank (2010) and FAO 33 (Doorenbos and Kassam, 1979). Average irrigation losses in the Zambezi area are estimated at 55% between gravity and sprinkler irrigation systems, considering conveyance, distribution and application losses (World Bank, 2010). Return flow are estimated at 30% for all cultures and catchments. Cultivation costs per hectare for different cultures are derived from Social Accounting Matrices of Malawi, Mozambique and Tanzania (IFPRI, 2014, 2015, 2017a). Cultivation costs include seeds, fertilizers, chemicals, labour and capital costs, the cost per hectare is calculated by dividing the total economic flow by the area cultivated the corresponding year. As few data are available, we consider a different cost per culture but use an average cost over all countries. The land costs are not included as the model internally represents a market for agricultural land use. We consider two farming zones per catchment, representing irrigated and rainfed land. Available area for rainfed and irrigated agriculture is obtained from the spatial data of the SPAM model (IFPRI and IIASA, 2017) and from World Bank (2010). For irrigated agriculture the crop choice is constrained by the observed area for each culture, this is to avoid over production of very profitable cash crops and takes into account non-represented physical, socio-economic or political constraints.

**Table 6: Represented crops and their importance in the agricultural sector.** The modelled crops represent more than 90% of the crop production, cultivated and irrigated area and of the production value. The production value excludes meat and dairy products. Some crops have a moderate impact on the global economy (e.g. cereals, rice and stimulants) but are important for irrigated agriculture. The share of irrigated area is from World Bank (2010), other indicators from FAO (2018) averaged over 2010 to 2016.

| Crop group | Main crops | Production | Cultivated area | Irrigated area | Production value |
|---|---|---|---|---|---|
| cassava | cassava | 22% | 7% | 0% | 30% |
| maize | maize | 20% | 43% | 7% | 18% |
| roots | potatoes and sweet potatoes | 9% | 3% | 0% | 18% |
| fruits | bananas, pineapples and coconuts | 5% | 2% | 3% | 5% |
| oilseeds | groundnuts, soybeans and sunflower | 3% | 13% | 5% | 7% |
| pulses | beans, peas and other pulses | 2% | 12% | 0% | 7% |
| cereals | wheat, sorghum, millet and barley | 2% | 7% | 17% | 2% |
| rice | rice | 1% | 1% | 13% | 2% |





| | | | | | |
|---|---|---|---|---|---|
| vegetables | tomatoes and other vegetables | 2% | 1% | 5% | 3% |
| stimulants | tobacco, tea and coffee | 1% | 2% | 8% | 7% |
| sugarcane | sugarcane | 28% | 1% | 33% | - |
| **TOTAL** | | **95%** | **92%** | **91%** | **99%** |

### 3.3.2 Crop markets, demands and values

To represent demand for crops, we consider one crop market per country, as data is usually at national level. Demand per country is derived from the "food supply quantity" data (in crops primary equivalent) from FAO (2018) averaging the years 2010-2016. National data is then downscaled by the ratio of population within the Zambezi basin to get the local demand. As no data was available, we assume the demand to be perfectly inelastic. To represent imports and exports out of the Zambezi area, we also consider an international market that has an infinite demand for cash crops like sugarcane and stimulants. Willingness to pay for crops in each crop market is evaluated as the "value of agricultural production" divided by the "production quantity" from FAO (2018). International market crop prices are the average import price for the Southern African market, calculated as the "value of import" divided by "import quantity" from FAO (2018). The same value is used for external supply costs from the international market, meaning that crop markets in the Zambezi can import crops at this price. As few data are available on transport and transaction costs, we assume that the transaction costs for imports from the international market are the difference between the international market price and the observed import price in each country from FAO (2018). Similarly, for exports towards the international market, the transaction costs are estimated as the difference between the international market price and the observed export price in each country. Transaction costs among countries are set as the difference between the import prices.



### 3.4 Development plans

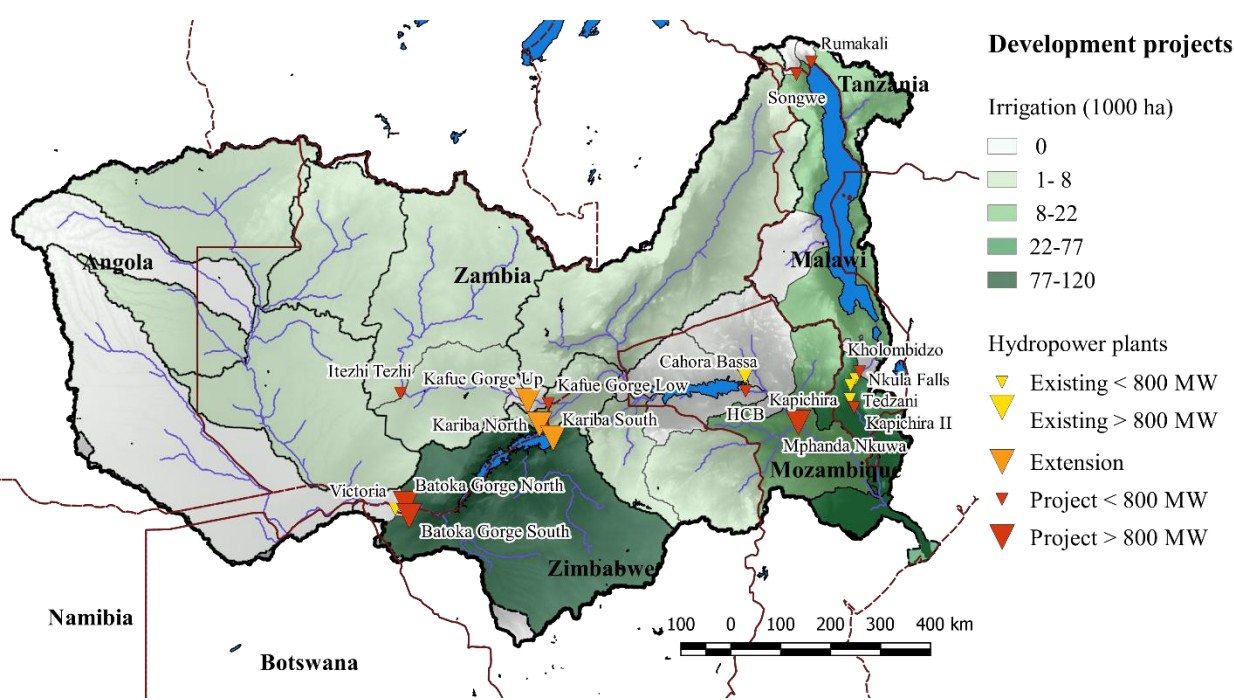

**Figure 5: Hydropower and irrigation development projects.** The major irrigation development projects are located at the Kariba Lake (Zimbabwe), in the Zambezi Delta (Mozambique) and in the Lower Shire (Malawi). The major hydropower projects are Batoka Gorge North and South with 800 MW in Zambia and 800 MW in Zimbabwe and Mphanda Nkuwa with 1300 MW in Mozambique.

### 3.4.1 Hydropower development plan

To respond to the rapidly increasing demand, SAPP countries are planning new or refurbished hydropower and thermal power plants, as well as new transmission lines. We refer to the " hydropower development plan" or "HDP" as the ensemble of projects described in World Bank (2010), it includes 15 projects with 7.2 GW of new operating capacity (Figure 5, Table 7). Investment costs in the hydropower projects range from 837 $ kW$^{-1}$ for Kapichira II to 3375 $ kW$^{-1}$ for the Batoka Gorge South project, total investment costs reach 12.5 billion dollars and fixed annual operating costs are estimated at 56 million dollars (IRENA, 2013). Transmission line projects are not considered as part of the HDP but are considered in future scenarios. Other power generation projects are not considered individually, however the representation of generic power technologies simulates additional investments in power generation.

**Table 7: Hydropower development projects.** For extension projects the original and projected capacity are indicated. Songwe (I+II+III) is an aggregation of three cascade hydropower projects. Projects are from World Bank (2010) and investment costs from IRENA (2013).

| Hydropower projects | Country | Capacity (MW) | Investment costs (M$) |
|---|---|---|---|
| Kapichira II | Malawi | 64 | 54 |
| Songwe (I+II+III) | Malawi | 340 | 456 |
| Kholombidzo | Malawi | 240 | 419 |





| | | | |
|---|---|---|---|
| Mphanda Nkuwa | Mozambique | 1300 | 2142 |
| HCB | Mozambique | 850 | 826 |
| Rumakali | Tanzania | 222 | 553 |
| Batoka Gorge North | Zambia | 800 | 2143 |
| Batoka Gorge South | Zimbabwe | 800 | 2700 |
| Kariba North | Zambia | 720->1200 | 643 |
| Kariba South | Zimbabwe | 750->1050 | 400 |
| KafueGorge Low | Zambia | 750 | 1607 |
| KafueGorge Up | Zambia | 990->1140 | 321 |
| Itezhi Tezhi | Zambia | 120 | 268 |

### 3.4.2 Irrigation development plan

We consider the irrigation development projects formulated in World Bank (2010), referred as "Irrigation development plan" or "IDP". With almost 100 identified irrigation projects aggregated per catchment, the IDP adds around 336 000 ha of equipped area to the 182 000 existing (Figure 5, Table 8). "Equipped area" refers to the actual land use, while "irrigated area" usually

5   double counts winter and summer crops on the same land. The total investment costs of the IDP are evaluated at 2.5 billion dollars (World Bank, 2010). The most important cultures in terms of area are: sugarcane (23%), rice (17%), wheat (15%) and maize (14%). The crop choice for the irrigated areas is constrained to the planned crops using data in World Bank (2010). We assume that irrigation projects replace existing rainfed areas as long as the irrigated area does not exceed the total available area.

10   **Table 8: Irrigation development projects.** The irrigation development projects are aggregated per catchment. Areas are expressed in terms of "equipped area" which counts the land use.

| Catchment | Existing area (1000 ha) | Project area (1000 ha) | Major culture |
|---|---|---|---|
| Kabompo | 0 | 6 | Wheat |
| Upper Zambezi | 3 | 5 | Sugarcane |
| Lungue | 1 | 1 | Rice |
| Luanginga | 1 | 5 | Rice |
| Baroste | 0 | 7 | Vegetables |
| Cuando | 1 | 0.3 | Rice |
| Kafue | 36 | 8 | Sugarcane |
| Kariba | 28 | 106 | Wheat |
| Luangwa | 10 | 6 | Wheat |
| Mupata | 14 | 6 | Stimulants |
| Lake Malawi (TAZ) | 12 | 12 | Rice |
| Lake Malawi (MLW) | 14 | 10 | Maize |
| Tete | 0 | 19 | Maize |
| Delta | 7 | 77 | Sugarcane |
| Kariba (BOT) | 0 | 14 | Maize |
| Lower Shire | 17 | 38 | Maize |
| Kafue Up | 4 | 6 | Soybeans |
| Harare | 22 | 8 | Wheat |
| Mazowe | 13 | 4 | Wheat |
| TOTAL | **183** | **336** | - |



### 3.5 Climate change, future scenarios and uncertainty analysis

The Zambezi river basin was classified by IPCC as being severely threatened by the potential effects of climate change (IPCC, 2001), according to World Bank (2010) the runoff might be reduced by 12 to 34 % depending on the regions. Furthermore, population is expected to grow from 40 to 70 million until 2050 (SADC et al., 2015). This will drastically increase energy and food demand and accentuate the pressure on ecosystems. As the investment plans involve infrastructure with a long lifetime, over 50 years for hydropower plants, it is crucial to consider the potential future climate and socio-economic scenarios. Table 9 offers an overview of the considered scenarios.

We consider four climate change scenarios from Cervigni et al. (2015): dry, semi-dry, semi-wet and wet for the period 2001 to 2050. The scenarios are derived using Bias Correction and Spatial Downscaling from the General Circulation Models (GCM) of the Climate Model Intercomparison Project – Phase 5 (Brekke et al., 2014), applied to historical climate data. Figure 6 shows how the different climate change scenarios impact the average evapotranspiration, precipitation and runoff in the Zambezi river basin. Like in World Bank (2010), we consider different flood restoration policies in the Zambezi delta: 4 500, 7 000 and 10 000 m³ s$^{-1}$ in February as sub-scenarios of the 2030 scenario.

Expected energy demand growth rates range from 0.7 % (Zambia) to 5.1 % (Tanzania) per year in the coming decades (SAPP, 2015), meaning that demand will more than double in some countries towards 2030. We consider a continuous growth rate of the demand for scenarios 2030 and 2050. Carbon pricing policies might have an important impact on energy generation, IRENA (2013) uses a carbon tax of 25 \$ t-$CO_{2eq}^{-1}$ in 2030. Thus, fuel prices would increase drastically: coal prices would double, while gas prices would increase by 30% (IRENA, 2013). We consider the expected 25\$/t-$CO_{2eq}$ carbon price for scenarios 2030 and 2050, and measure the sensitivity of this policy by varying the carbon tax from 0 to 50 \$ t-$CO_{2eq}^{-1}$ in the 2030 scenario. Capital costs of solar photovoltaic are expected to be halved until 2030 (IRENA, 2013), we consider a capital cost of 1000 \$ kW$^{-1}$ in scenarios 2030 and 2050, and vary it from 2000 \$ kW$^{-1}$ to 500 \$ kW$^{-1}$ in the 2030 scenario. Future transmission lines, between Malawi and Mozambique, Tanzania and Zambia and Namibia and Angola (SAPP, 2015) are considered as constructed in the 2030 and 2050 scenarios.

Crop demand is expected to increase by 10% (roots and tuber, Angola) to 140% (fruits and vegetables, Zambia) by 2030, depending on crops and countries (IFPRI, 2017b). We consider demand growth in the 2030 and 2050 scenarios, using projected demands for 2030 and 2050 from IFPRI (2017b). According to OECD and FAO (2017), yields will increase by 0.5 % (roots, Mozambique) to 3.8 % (rice, Zambia) per year; we consider this in the 2030 and 2050 scenarios, assuming continuous growth. This might be optimistic when FAO (2018) data shows that yields for some crops (e.g. rice, wheat, and sugarcane) in the Zambezi countries have been stable for the past 20 years. Thus, we also consider no yield growth for the sensitivity analysis of the 2030 scenario. National and crop specific yield data are available for Mozambique, Tanzania and Zambia, the sub-Saharan average is used for the other countries. Similarly, rainfed area should increase by 1.2% (Tanzania) to 2% (Mozambique) per year (OECD and FAO, 2017), we include these changes in the 2030 and 2050 scenarios. As no data was



available, we assume world market crop prices remain stable in the future scenarios. However, we test the sensitivity of this

assumption by varying world market crop prices by +/- 20% in the 2030 scenario.

**Table 9: Main scenarios.** The table presents trends in the water, energy and agriculture sectors for the three main scenarios: 2010, 2030 and 2050. The sub-scenarios are relative to the 2030 scenario, to evaluate the sensitivity of the results to climate change, world market crop prices, $CO_2$ pricing policies, capital costs of solar photovoltaic capacity, and environmental flow policy. [1]The price variation is only for the world market. [2]Flood level restoration at the Zambezi delta during the month of February.

| Scenarios | 2010 | 2030 | 2050 | Sub-scenarios of 2030 | Source |
|---|---|---|---|---|---|
| Crop demand (Mt yr$^{-1}$) | 26 | +60% | +144% | - | (FAO, 2018; IFPRI, 2017b) |
| Cultivated Area (M ha$^{-1}$) | 6.6 | +39% | +92% | - | (OECD and FAO, 2017) |
| Yields (t ha$^{-1}$) | - | +41% | +100% | - | (OECD and FAO, 2017) |
| Crop Value ($ t$^{-1}$) | 669 | - | - | -20% to +20%[1] | (FAO, 2018) |
| Energy demand (GWh yr$^{-1}$) | 68 338 | +87% | +278% | - | (SAPP, 2015) |
| $CO_2$ price ($ t-$CO_{2eq}$^{-1}$) | 0 | 25 | 25 | 0 to 50 | (IRENA, 2013) |
| Solar capital costs ($ kW$^{-1}$) | 2 000 | 1000 | 1000 | 2000 to 500 | (IRENA, 2013) |
| Runoff (Mm³ yr$^{-1}$) | 114 868 | - | - | -54% to +35% | (Cervigni et al., 2015) |
| Flood level[2] (m³ s$^{-1}$) | 0 | - | - | 4 500 to 10 000 | (World Bank, 2010) |



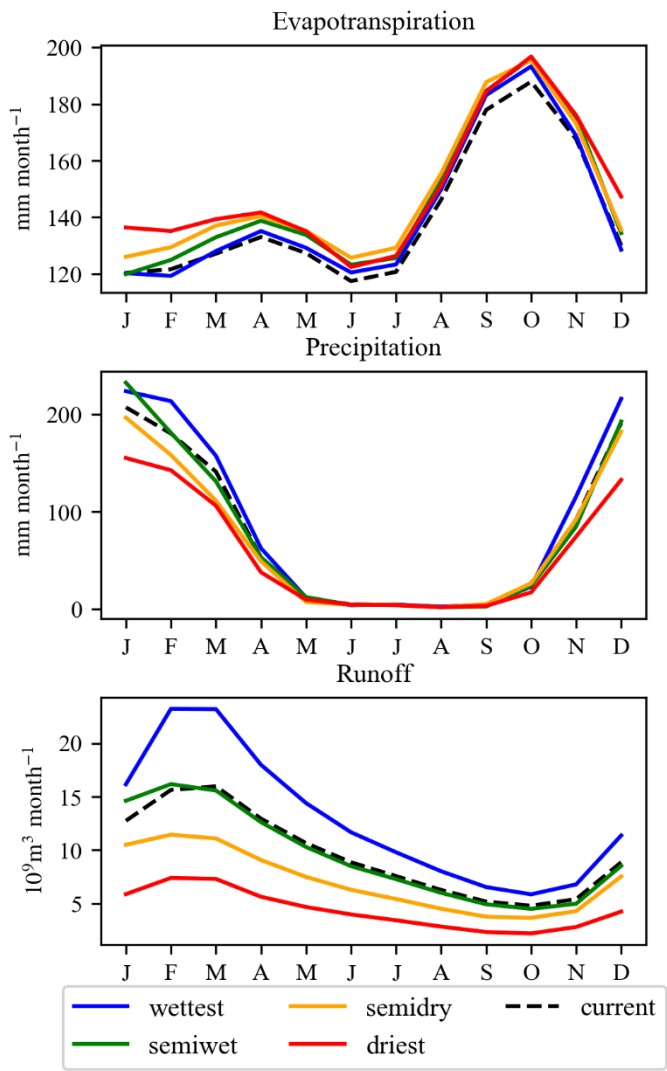

**Figure 6: Impact of climate change on hydrologic parameters.** The average yearly pattern of evapotranspiration, precipitation and runoff is shown for the four climate change scenarios and the current climate.

## 4. Results and discussion

5    In this section, we illustrate how the Zambezi model can be used to answer questions such as "What are the potential impacts of climate change on the agriculture and energy systems?", "What are the benefits of the hydropower and agricultural development plans?", "What is the sensitivity of these benefits regarding uncertainties in policies, future climate and socio-



economic development ?", "What are the synergies and trade-offs between the irrigation and hydropower development plan?", and "What are the opportunity costs of restoring flood regimes in the Zambezi delta ?"

## 4.1 Model validation

To validate the model, we show the overall results of the 2010 scenario, which is the situation in the Zambezi river basin
around the year 2010. The water balance for the Zambezi river basin (Table 10) shows that the main water consumption is evaporative losses from reservoirs (mainly the Kariba and Cahora Bassa dams). The total water consumption reaches around 10% of the total available water. Agricultural uses represent only around 25 % of the total water consumption, and domestic and industrial consumption around 5%. The average runoff and reservoir evaporation vary significantly among the different studies (Table 10), it is unclear if the studies report net or gross evaporation (including or excluding rainfall on the reservoir
area). For the average runoff, the difference is most likely due to different reference periods, according to our data the average yearly runoff from 1960 to 1980 was 129 000 $10^6$ m³, while from 1980 to 2000 it was only 100 600 $10^6$ m³.

The model reproduces the patterns of agricultural water consumption per country (Table 11), with some differences between Zambia and Zimbabwe. These differences may be explained by differences in aggregation of agricultural areas at the border between Zambia and Zimbabwe between this and the World Bank (2010) study. The modelled value of crop production (Table
12) is in accordance with observations; main production is in Malawi, followed by Zambia and Zimbabwe. The "observed" value is obtained by downscaling national statistics assuming constant per capita value, and is therefore not a precise number. Hydroeconomic models often valuate agriculture by considering a willingness-to-pay for water by agriculture users, or by representing crop production and valuating crops at the farm level (Bauer-Gottwein et al., 2017; Harou et al., 2009). Non-market approaches include the following limitations: (1) crop demand is not represented, (2) crop trade, transaction costs, and
losses are not represented, (3) food security constraints cannot be represented, (4) the interactions with rainfed agriculture cannot be represented, and (5) it requires to calculate an exogenous willingness-to-pay for water or crops for each considered socio-economic development scenario potentially affecting the crop markets. In this study we use a market approach valuating crops at the consumer level. Non-market and market approaches can be similar if irrigated crops are a marginal share of the total production, and if supply, demand, and trade of crops remain stable. In the Zambezi, irrigated crops are a small share of
the total production. However, we analyse the potential impacts of climate change, significantly affecting rainfed production, and therefore crop supply and food security constraints, in the context of an increasing crop demand. Therefore, the market approach is necessary to perforn the analysis of this study.

Hydropower production per plant (Table 13) is similar to World Bank (2010), with small variations linked to differences in modelling approach and hydrologic data. A common approach to value hydropower production, is to use the concepts of firm
and secondary power and value them differently to indirectly represent the power market. In the World Bank (2010) study, firm power is calculated as the power production available 99% of the time (at the monthly scale) at a single plant, while secondary power is the remaining power production. Indeed, assuming a constant power demand at the monthly time scale, firm power can replace investments in thermal power capacity, while secondary power needs to be balanced by thermal





capacity. Thus, secondary power is only saving fuel costs but not ramping and capacity investment costs and has therefore a lower value. In this study we do not use the firm power concept but simulate the power market instead. Although the hydropower plant production is not optimized for firm power, we find similar results to the World Bank (2010) study (Table 13). The reasons are that: (1) we do not consider seasonal variations in the availability of a power source (e.g. solar capacity has a diurnal variation but seasonal variation is assumed to be negligible), (2) low and high flow seasons occur at the same time of the year throughout the basin reducing the benefit of coordinating hydropower production in different subbasins, and (3) reservoirs have a high storage capacity enabling hydropower plants to operate with relatively stable monthly releases. Although the firm power and market approaches give similar results in this case, the firm power approach has some limitations: (1) it does not represent transmission constraints which are considered to be important in the SAPP power system (Spalding-Fecher et al., 2017b), (2) it does not consider the power demand, (3) it does not enable coordination between several hydropower plants to balance fluctuations in production at individual plants, (4) it does not enable representation of the benefits of hydropower as a peak power source, satisfying peak demand or balancing an unstable power source such as solar or wind, and (5) it cannot be used to evaluate the impact of carbon tax policies, capital costs of renewable technologies, and future energy demand, which would require an exogenous model to calculate firm and secondary power values for each scenario.

In general, the model reproduces the trends observed in the reference scenario for the water, energy and agriculture systems, but some differences appear. Therefore, in the following analysis, most of results are not shown as absolute values, but as relative changes between different scenarios.

**Table 10: Water balance of the reference scenario.** Results are presented as the average for the 40 years simulation. The amount of runoff and reservoir evaporation varies significantly depending on the studies. Average yearly runoff might be influenced by the historical period considered. *It is not clear if the cited studies report reservoir evaporation as net (including rainfall) or gross values, this might explain differences. **The publication reports an average runoff of 130 300 $10^6\,m^3\,year^{-1}$, however this is believed to be a reporting mistake (Strzepek 2019, Personal communication), the average runoff used in the calculations is 107 000 $10^6\,m^3\,year^{-1}$.

| Water Balance [$10^6\,m^3\,year^{-1}$] | This study | World Bank (2010) | Tilmant et al. (2012) | (Beilfuss, 2012) | (Euroconsult and Mott MacDonald, 2008) |
|---|---|---|---|---|---|
| Runoff | 114 868 | 107 000** | | 110 732 | 103 224 |
| Domestic and Industrial consumption | 772 | 797 | | | 344 |
| Agricultural consumption | 3 409 | 3 234 | | | 1 478 |
| Net Reservoir Evaporation* | 8 825 | 8 000 | 7 800 | 12 181 | 16 989 |

**Table 11: Agricultural water consumption in the Zambezi river basin.**

| Agricultural water consumption ($10^6\,m^3\,yr^{-1}$) | World Bank (2010) | This study |
|---|---|---|
| Angola | 75 | 119 |
| Botswana | 0 | 0 |
| Malawi | 494 | 575 |
| Mozambique | 134 | 117 |



| | | |
|---|---|---|
| Namibia | 2 | 0 |
| Tanzania | 154 | 102 |
| Zambia | 879 | 1 353 |
| Zimbabwe | 1 496 | 1 153 |
| **Total** | **3 234** | **3 419** |

**Table 12: Value of crop production in the Zambezi river basin.** Data from FAO (2018) is at national level, downscaled to the Zambezi river basin assuming constant per capita value, therefore it might not be fully representative for the actual regional value.

| Value of crop Production (M$ yr$^{-1}$) | (FAO, 2018) | This study |
|---|---|---|
| Angola | 207 | 216 |
| Botswana | 0 | 1 |
| Malawi | 4 497 | 4495 |
| Mozambique | 418 | 329 |
| Namibia | 11 | 0 |
| Tanzania | 150 | 241 |
| Zambia | 1 082 | 1 325 |
| Zimbabwe | 572 | 527 |
| **Total** | **6 936** | **7 130** |

5 **Table 13: Hydropower production**. Simulated hydropower production is compared to the results of the MSIOA study (World Bank, 2010), Firm power is calculated as the power production available 99% of the time (at the monthly scale) at a single plant.

| Hydropower Production (GWh yr$^{-1}$) | World Bank (2010) | | This study | |
|---|---|---|---|---|
| | Total | Firm | Total | Firm |
| Cahora Bassa | 13 535 | 11 922 | 12 541 | 9 232 |
| Kafue Gorge Up | 6 785 | 4 695 | 7 498 | 4 857 |
| Kapichira | 560 | 455 | 421 | 275 |
| Kariba North | 3 834 | 3 184 | 4 409 | 2 809 |
| Kariba South | 3 834 | 3 184 | 4 659 | 2 926 |
| Nkula | 1 017 | 462 | 869 | 711 |
| Tedzani | 722 | 300 | 597 | 455 |
| Victoria Falls | - | - | 852 | 852 |
| **Total** | **30 247** | **22 776** | **31 848** | **22 116** |

## 4.2 Potential impacts of climate change

**Table 14: Impact of climate change on the water-energy-food system.** The results show the difference in key indicators with and without the four climate change scenarios for the 2030 scenario without infrastructure development plans.

| Key indicators | Driest | Semi dry | Semi wet | Wettest |
|---|---|---|---|---|
| Available runoff (10$^6$m$^3$ yr$^{-1}$) | -62 195 | -29 838 | -892 | 40 156 |
| Agriculture sector benefits (M$ yr$^{-1}$) | -1 644 (-15%) | -176 (-2%) | -206 (-2%) | 218 (+2%) |
| Crop price index | +33% | +7% | +4% | -4% |





| | | | | |
|---|---|---|---|---|
| Energy sector benefits (M$ yr$^{-1}$) | -714 | -365 | -24 | 187 |
| Hydropower production (GWh yr$^{-1}$) | -15 668 | -8 801 | -518 | 4 981 |
| CO2 emissions (Mt yr$^{-1}$) | 10 | 5 | 0 | -2 |

Climate change is found to have important potential impacts, inducing losses of more than 2.3 billion dollars per year in the driest scenario to increasing benefits by 400 million dollars per year in the wettest scenario (Table 14). In the driest scenario, average runoff is more than halved, reducing by 50% current hydropower production. This causes economic losses to the energy sector of more than 700 M$ yr$^{-1}$. In the wettest scenario, the average runoff increases by 35%, increasing hydropower production by almost 5 000 GWh yr$^{-1}$, resulting in an increased benefit of 220 M$ yr$^{-1}$ for the energy sector. The agricultural sector is particularly sensitive as it mainly relies on rainfed agriculture. The driest scenario seems to be a critical threshold where an important portion of rainfed cultures show low yields. Indeed, from the semi dry to the driest scenario the induced economic losses rise from 200 to 1 640 M$ yr$^{-1}$ and the crop price index from +4% to +33%. Similarly, the value of water in the Shire river (Malawi) is not affected in the semi dry scenario but rises considerably in the driest scenario (Figure 7). In fact, in the semi dry scenario Malawi observes losses of only 8 M$ yr$^{-1}$, but in the driest scenario, losses reach more than 800 M$ yr$^{-1}$ (mostly to the agriculture sector).

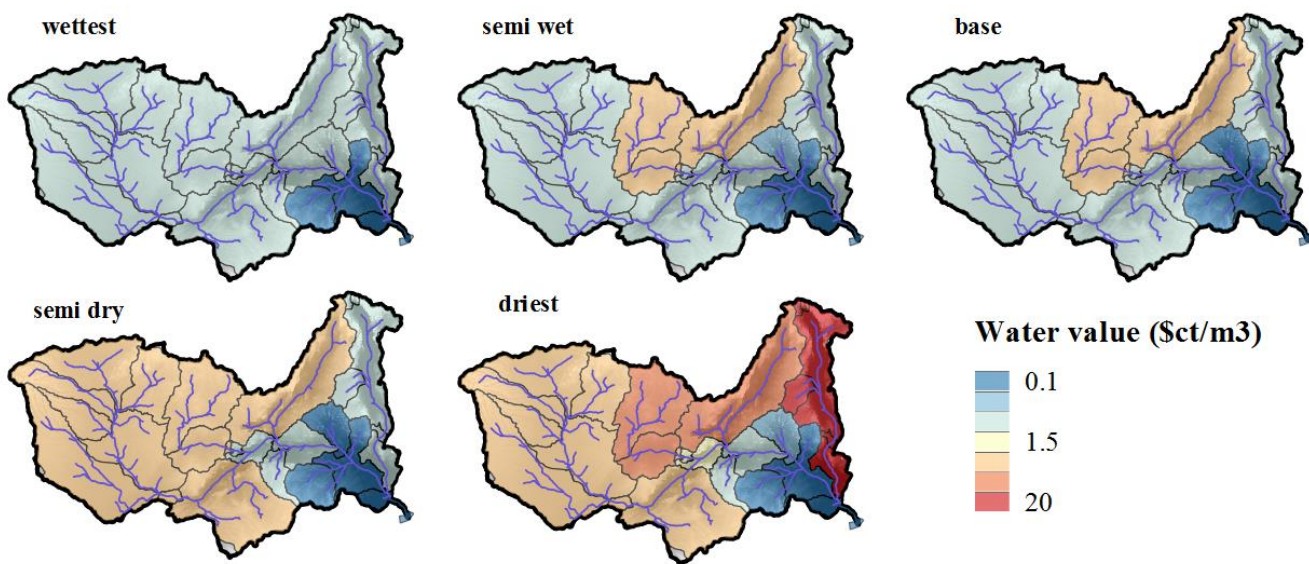

**Figure 7: Average water value under climate change scenarios.** The "water value" is the result of the shadowprices/duals of the constraint of the hydroeconomic model, it represents the potential economic benefits of an additional m³ of water in the river. The increased water value is a sign for increased water scarcity.

## 4.3 Hydropower development plan

The hydropower development plan (HDP) is found to produce an extra 28 000 to 33 000 GWh yr$^{-1}$ (Table 15), which is in accordance with World Bank (2010) (30 400 GWh yr$^{-1}$). For the 2030 scenario, the resulting value is around 1 932 M$ yr$^{-1}$. Considering the total investment costs of 12.5 billion dollars as well as the fix annual operating costs and a lifetime of 50 years




for the hydropower projects, this represents an internal rate of return of 14.7 % (assuming overnight construction of the hydropower projects, excluding cost and benefits linked to non-represented elements such as fishing, tourism, flood regulation, navigation and ecosystem services). The main benefits occur in countries with new major hydropower projects (Zambia, Zimbabwe, Mozambique and Malawi), however even countries that do not have any projects (e.g. Namibia) benefit from

cheaper power imports (Table 16). While Zambia and Zimbabwe use most of the additional energy for own supply, Malawi exports half and Mozambique exports all of it. The impact of the HDP on the electricity price is relatively small as an important share of the demand is satisfied through thermal power. Therefore, the economic impact is mainly producer surplus, while consumer surplus is limited (Table 15). However, this varies locally, in Malawi the HDP makes the country almost independent from thermal power, lowering electricity prices by 16 $ MWh$^{-1}$ and generating important consumer surplus (Table 16).

Hydropower production is around 4 000 GWh yr$^{-1}$ lower in the 2010 than in the 2030 scenario (Table 15), this can be explained by demand limits and the fact that in the 2010 scenario Malawi is not connected to the SAPP grid and cannot export over production of its Run-Off-the-River hydropower plants. In fact, under the HDP in the 2030 scenario, power transmission among SAPP countries is considerably increased, including transfers from Malawi to Mozambique (Figure 8). In the practical implementation of the HDP, projects would be implemented gradually, therefore the demand constraint would probably not

be a problem, however in this study we do not consider the timing and sequencing of the projects. The main difference between the 2010 and the 2030 scenarios is the generated benefits. The driving factor is the potential carbon pricing policy, as it will considerably affect fuel costs and therefore the cost of generating power with thermal power plants (Figure 9). We consider carbon price as a cost, while it could also be considered as a tax and therefore have no effect on the welfare impact (being a money transfer from producers to states). However, the principle of a carbon price/tax is to compensate for CO$_2$ emissions,

which will therefore result in a cost. Thus, the price of electricity for consumers increases from 53 $ MWh$^{-1}$ in the 2010 scenario to 73 $ MWh$^{-1}$ in the 2030 scenario, increasing the value of developing hydropower.

The HDP has no impact on the agricultural system (Table 15), neither positive or negative, and vice versa, the development of the irrigation development plan does almost not affect its value (Figure 9). The value of the HDP is relatively sensitive to climate change; it varies from 1 651 to 2 075 M$ yr$^{-1}$ for the driest to the wettest scenarios. The additional hydropower

production is severely impacted in the driest scenario, producing only 25 000 GWh yr$^{-1}$ against 37 000 GWh yr$^{-1}$ for the wettest scenario. However, climate change has more impact on the current hydropower plants, where the driest climate change scenario is found to halve current power production (Table 14). Another influencing parameter for the value of the HDP is the capital costs of solar photovoltaic power, as this affects the cost of producing alternative energy. With solar capital costs ranging from 2000 $ kW$^{-1}$ to 500 $ kW$^{-1}$, the electricity consumer price varies from 80 $ MWh$^{-1}$ to 70 $ MWh$^{-1}$, and the value of the HDP

from 2070 to 1880 M$ yr$^{-1}$. Excluding solar photovoltaic technology from the simulation does result in the same value for the HDP in the 2030 scenario. Solar power has a double effect on the value of hydropower plants: on one hand, it reduces electricity prices and therefore the value of hydropower energy; on the other hand, it increases intermittency of the power system and so the value of flexible hydropower generation. As there is already an important hydropower capacity available in the Zambezi river basin, both effects compensate in this case.





**Table 15: Impacts of the Hydropower development plan.** Results are shown as the difference with and without the hydropower development plan for the 2010, 2030 and 2050 scenarios. Sector benefits exclude construction costs.

| Impacts of the Hydropower development | 2010 | 2030 | 2050 |
|---|---|---|---|
| Energy sector benefits (M$ yr$^{-1}$) | 1 042 | 1 932 | 2 058 |
| Agriculture sector benefits (M$ yr$^{-1}$) | 0 | 1 | 1 |
| Energy consumer surplus (M$ yr$^{-1}$) | 737 | 271 | 206 |
| Energy producer surplus (M$ yr$^{-1}$) | 305 | 1 661 | 1 852 |
| Reservoir evaporation ($10^6$m$^3$ yr$^{-1}$) | 288 | -132 | -81 |
| Water shadowprice ($ $10^3$m$^{-3}$) | 3 | 8 | 10 |
| Hydropower production (GWh yr$^{-1}$) | 28 536 (+95%) | 32 809 (+100%) | 32 360 (+100%) |
| Energy trade (GWh yr$^{-1}$) | 15 767 (+195%) | 14 238 (+185%) | 6 570 (+150%) |
| Thermal power investments (MW) | -3 651 | -4 859 | -5 657 |
| Solar power investments (MW) | 0 | -3 387 | -2 955 |
| Energy price ($ MWh$^{-1}$) | -12 (-15%) | -3 (-5%) | -2 (-3%) |
| CO$_2$ emissions (Mt yr$^{-1}$) | -25 | -23 | -24 |

**Table 16: Country-scale impacts of the Hydropower development plan.** Results are shown as the difference with and without the hydropower development plan for the 2030 scenario. The added value excludes construction costs.

| Impacts of the Hydropower development | Angola | Botswana | Malawi | Mozambique | Namibia | Tanzania | Zambia | Zimbabwe |
|---|---|---|---|---|---|---|---|---|
| Added value (M$ yr$^{-1}$) | 0 | 1 | 196 | 470 | 12 | 128 | 654 | 467 |
| Hydropower production (GWh yr$^{-1}$) | 0 | 0 | 3 792 | 9 321 | 0 | 1421 | 11 077 | 7 197 |
| Energy exports (GWh yr$^{-1}$) | 0 | 1 | 1 635 | 9 604 | 0 | 58 | 1 549 | 1 391 |
| Thermal power investments (MW) | 0 | 0 | -438 | 0 | 0 | -200 | -2 064 | -1 418 |
| Solar power investments (MW) | 0 | 0 | -455 | -1 435 | 0 | 0 | -1 051 | -446 |
| Energy consumer surplus (M$ yr$^{-1}$) | 0 | 0 | 69 | 45 | 0 | 0 | 104 | 52 |
| Energy producer surplus (M$ yr$^{-1}$) | 0 | 1 | 127 | 425 | 12 | 128 | 550 | 415 |
| Energy price ($ MWh$^{-1}$) | 0 | 0 | -16 | -5 | 0 | 0 | -4 | -2 |
| CO$_2$ emissions (Mt yr$^{-1}$) | 0 | -1 | -1 | 0 | -1 | -1 | -7 | -6 |



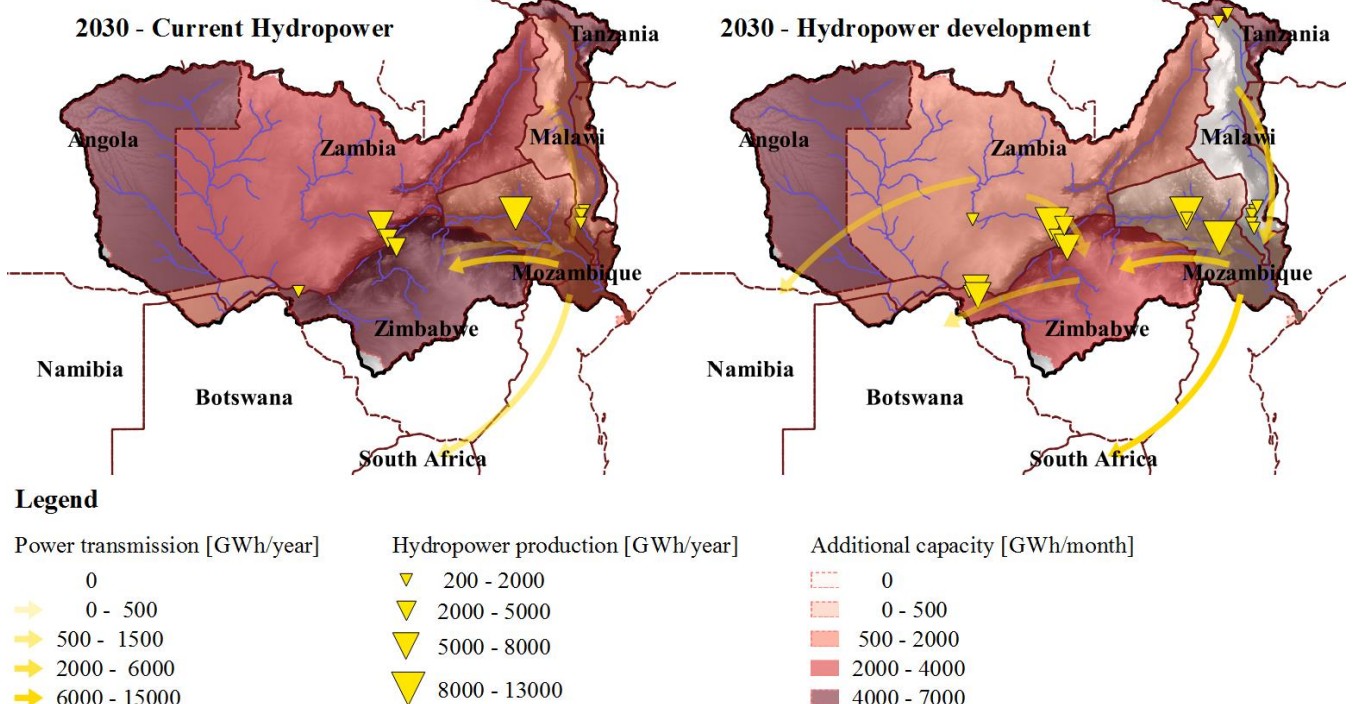

**Figure 8: Power transmission, hydropower production and additional capacity investments before and after implementation of the hydropower investment plan.** The implementation of the hydropower development plan is found to increase considerably the power exchanges among the SAPP countries and reduce the needs to invest in additional power generation capacity.



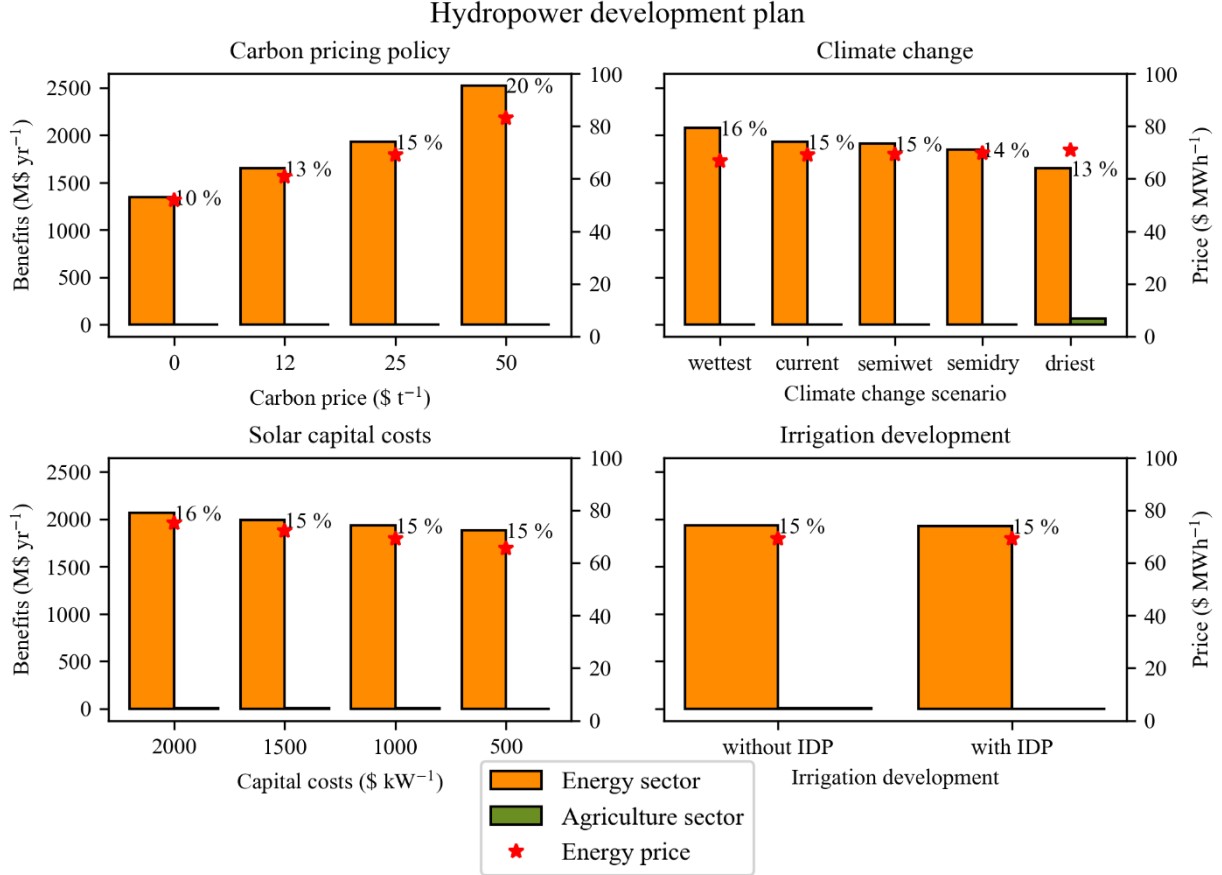

**Figure 9: Sensitivity of the hydropower development plan benefits to uncertainties in future climate and socio-economic development.** Benefits are calculated from with/without analyses of the hydropower development in variations of the 2030 scenario, percentages indicate the internal rate of return. The value of the hydropower development plan is found to be most sensitive to carbon pricing policy, climate change and capital costs of solar photovoltaic. Current carbon price is 0$/t and 25$/t is used in the 2030 scenario; current solar capital costs are 2000 $ kW$^{-1}$ and 1000 $ kW$^{-1}$ is used in the 2030 scenario.

## 4.3 Irrigation development plan

The irrigation development plan (IDP) is valued between 650 and 1220 M$ yr$^{-1}$ depending on the scenario (Table 17). Considering investment costs of 2.5 billion dollars and a lifetime of 20 years for the infrastructure, this corresponds to an internal rate of return of 26 % to 49 % (ignoring maintenance costs). The important variation can be explained by the expected growth in yields that should more than double between the 2010 and 2050 scenarios (OECD and FAO, 2017). This assumption might be optimistic given that according to FAO (2018) data, for several crops yields have been relatively stable over the past twenty years. The implementation of the IDP more than doubles irrigated area (+430 000 ha), as well as water consumption (+5 200 10$^6$ m³ yr$^{-1}$). Because of the increased water consumption, the benefits of the IDP for the agriculture sector are to a limited extent counterbalanced by losses in the energy sector. In fact, around 5% of the benefits are lost because of resulting hydropower shortages of about 1 200 GWh yr$^{-1}$ (Table 17). This is a lower level of trade-offs than in World Bank (2010),



which estimated hydropower shortages around 2 700 GWh yr$^{-1}$. The most affected country is Mozambique (-650 GWh per year); because it is the most downstream country, its hydropower production is affected by the water consumption in Zimbabwe, Zambia and Malawi (Table 18, Figure 10). More than 80% of the value of the IDP is generated through crop trade (Table 17, Figure 10), thus world market crop prices are a very sensitive parameter (Figure 11). A reduction of 20 % in world

market crop prices would reduce by 25 % the value of the IDP. As most of the profits are linked to exports, the IDP has a relatively small impact on crop prices, and therefore, benefits occur mostly as producer surplus rather than consumer surplus (Table 17). A drier climate has a twofold impact on the IDP (Figure 11): it reduces rainfed production and thus increases the value of irrigation, but it also increases trade-offs with the energy sector. In fact, in the current climate scenario the IDP saves 48 M$ yr$^{-1}$ of import value from the world crop market to satisfy food security constraints, while in the driest scenario it saves

95 M$ yr$^{-1}$ of import value. This shows the importance of representing rainfed agriculture to assess the value of irrigation projects. However, hydropower shortages induced by additional water consumption range from 515 GWh yr$^{-1}$ in the wettest scenario to 1 600 GWh yr$^{-1}$ in the driest scenario, inducing losses in the range of 24 to 104 M$ yr$^{-1}$ (representing up to more than 10% of the benefits) which counterbalance the import substitution effect in the crop market. The trade-offs are limited because the water consumption is a small share of the available water (around 15% including the irrigation development).

Implementation of the hydropower development plan is not found to increase trade-offs between irrigation and hydropower and has no impact on the value of the IDP (Figure 11).

**Table 17: Impacts of the irrigation development plan.** Results are shown as the difference with and without the irrigation development plan for the 2010, 2030, and 2050 scenarios. Sector benefits exclude project development costs.

| Impacts of the Irrigation development plan | 2010 | 2030 | 2050 |
|---|---|---|---|
| Energy sector benefits (M$ yr$^{-1}$) | -29 | -51 | -50 |
| Agriculture sector benefits (M$ yr$^{-1}$) | 670 (+12%) | 948 (+9%) | 1 270 (+8%) |
| Crop consumer surplus (M$ yr$^{-1}$) | 150 | 186 | 119 |
| Crop producer surplus (M$ yr$^{-1}$) | 520 | 762 | 1151 |
| Gross irrigated area (1000 ha) | 420 (+200%) | 430 (+185%) | 400 (+178%) |
| Crop Price Index | -15% | -12% | -7% |
| Irrigation consumption ($10^6$m$^3$ yr$^{-1}$) | 5 151 (+155%) | 5 252 (+160%) | 5 052 (+154%) |
| Reservoir evaporation ($10^6$m$^3$ yr$^{-1}$) | 160 | -58 | -67 |
| Hydropower production (GWh yr$^{-1}$) | -1 219 | -1 231 | -1 187 |
| Thermal power investments (MW) | 15 | 0 | 0 |
| Solar power investments (MW) | 0 | 144 | 144 |

**Table 18: Country-scale impacts of the irrigation development plan.** Results are shown as the difference with and without the irrigation development plan for the 2030 scenario. The added value excludes construction costs.

| Impacts of the Irrigation development plan | Angola | Botswana | Malawi | Mozambique | Namibia | Tanzania | Zambia | Zimbabwe |
|---|---|---|---|---|---|---|---|---|



| | | | | | | | | |
|---|---|---|---|---|---|---|---|---|
| Added value (M\$ yr$^{-1}$) | 7 | 24 | 75 | 192 | 1 | 4 | 120 | 476 |
| Net irrigated area (1000 ha) | 9 | 13 | 43 | 95 | 0 | 10 | 33 | 121 |
| Crop consumer surplus (M\$ yr$^{-1}$) | 1 | 1 | 5 | 5 | 1 | 4 | 40 | 130 |
| Crop producer surplus (M\$ yr$^{-1}$) | 5 | 24 | 70 | 209 | 0 | 0 | 99 | 355 |
| Crop Price Index | -4% | -45% | 0% | 7% | -6% | -2% | 9% | -24% |
| Irrigation consumption ($10^6$m$^3$ yr$^{-1}$) | 178 | 221 | 411 | 1383 | 0 | 43 | 671 | 2345 |
| Hydropower production (GWh yr$^{-1}$) | 0 | 0 | -4 | -654 | 0 | 0 | -430 | -143 |
| Thermal power investments (MW) | 0 | 0 | 0 | 0 | 0 | 0 | 0 | 0 |
| Solar power investments (MW) | 0 | 0 | 0 | 0 | 0 | 0 | 144 | 0 |

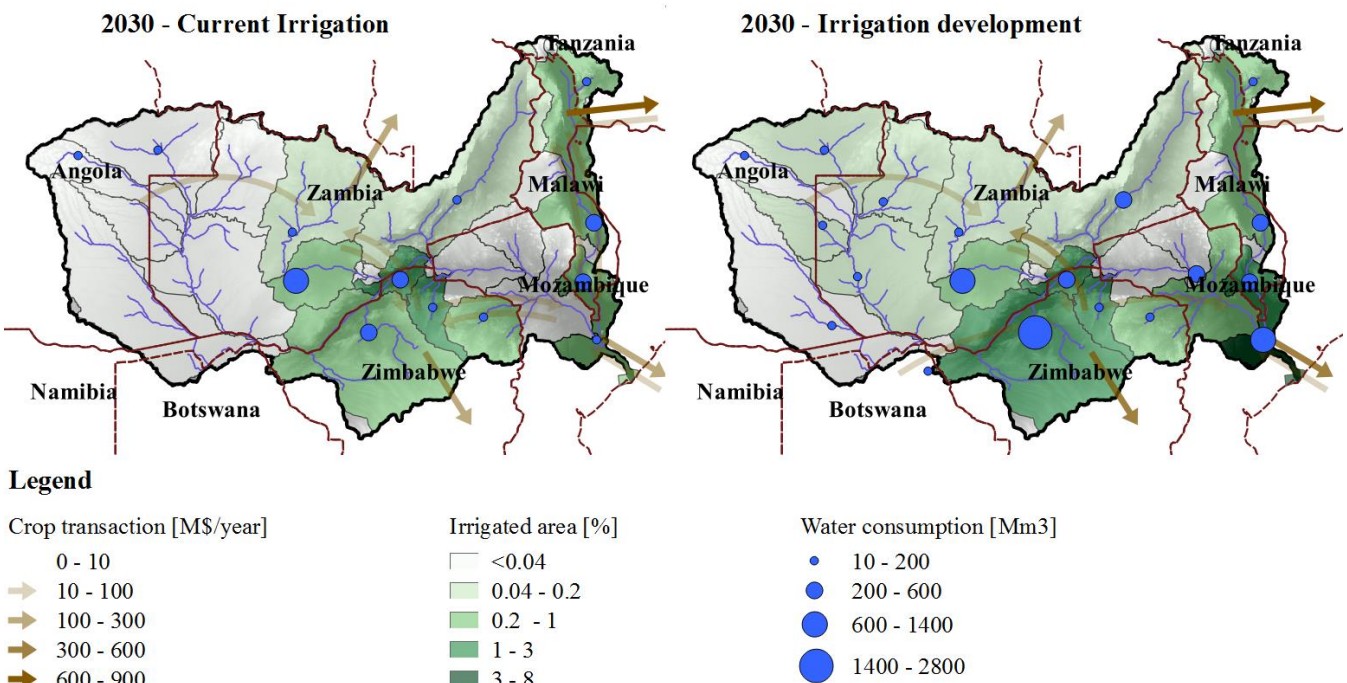

**Figure 10: Crop trade and irrigation water consumption before and after implementation of the irrigation development plan (IDP).**
Implementation of the IDP is found to increase crop exports to the world market and more than double water consumption. The "crop transactions" towards the exterior of the Zambezi area represent exports to the world market.





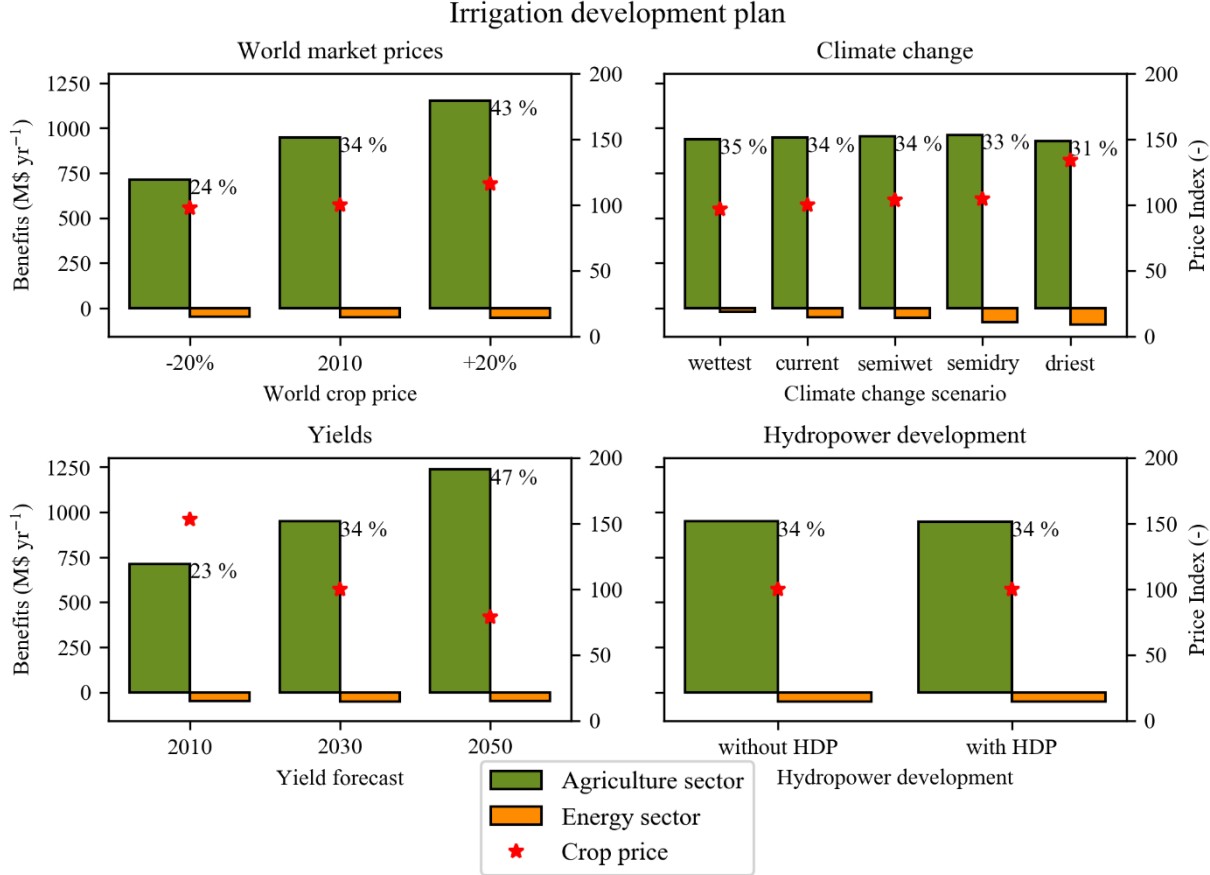

**Figure 11: Sensitivity of irrigation development plan to uncertainties in future climate and socio-economic development.** Benefits are calculated from with/without analyses of the irrigation development in variations of the 2030 scenario, the percentages show the internal rate of return. The value of the irrigation development plan is found to be very sensitive to crop yields, world market crop prices, and less to the climate change scenario. The crop price index is proportional to the weighted average crop price of supplied crops within the Zambezi, 100 is the reference value for the 2030 scenario.



## 4.4 Restoration of flood regimes

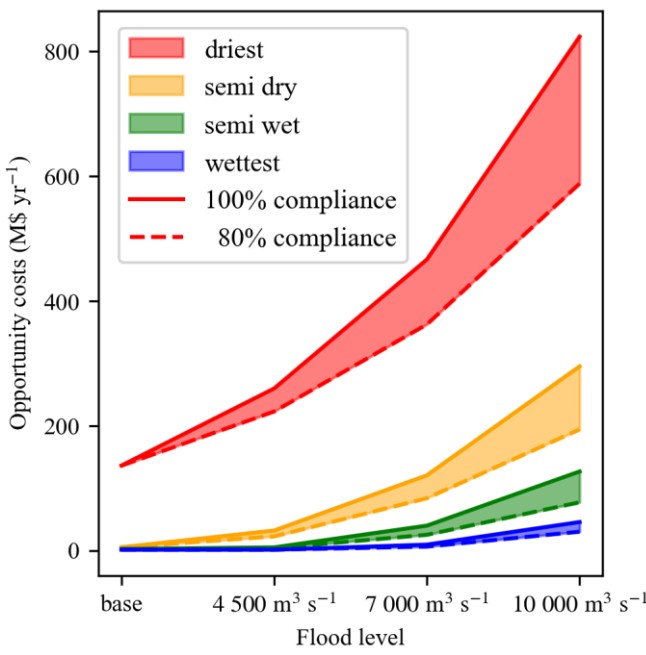

**Figure 12: Opportunity costs of restoring flooding regimes in the Zambezi delta.** The opportunity costs of the different flood level
targets are calculated for the four different climate scenarios assuming the hydropower and irrigation development plan are implemented.
The "base" flood level corresponds to the environmental flows constraint without including flood restoration. A 100 % compliance means
the flood level is ensured every year, while an 80% compliance means the environmental flow constraint must be achieved four out of five
years.

Natural flooding in the wetlands and the Zambezi delta was severely affected by the construction of the Kariba and Cahora

Bassa dams. Indeed, as at the monthly scale, thermal power plants have a stable production, hydropower production is more

valuable when it is as constant as possible, therefore the dams tend to stabilize the water releases throughout the year. However,

floods play an important role for ecosystems in the wetlands and therefore a potential policy is to restore the natural floods

(World Bank, 2010). Figure 12 shows the opportunity costs of restoring floods in February for three flood levels and the four

climate change scenarios, considering a 100 % (the constraint is fulfilled every year) and an 80 % compliance (the constraint

must be fulfilled 4 out of 5 years). Opportunity costs of the "base" environmental flow policy are almost zero except for the

driest climate change scenario. The restoration of the natural floods induces increasing costs with the flood level target: costs

reach up to more than 800 M\$ yr$^{-1}$ for the driest scenario and the highest flood level, but stay under 150 M\$ yr$^{-1}$ for the semi

wet and wettest scenarios. This is in accordance with Tilmant et al. (2012) who found opportunity costs of 104 M\$ yr$^{-1}$ for

restoring floods under current climate. We consider here only opportunity costs of the policy as trade-offs with hydropower

production and irrigation, but not benefits linked to direct and indirect use and non-use values of ecosystems or costs linked to

population displacement. More than half of the population depends directly on wetland ecosystems (SADC et al., 2015),





therefore benefits linked to the protection of ecosystems might be important and a complete cost-benefit analysis would reveal the value of such environmental policies.

## 5. Limitations and further research

By connecting the water, energy and food systems in a holistic framework and using an economic optimization approach we showed how we could evaluate the development plans in the Zambezi river basin considering different scenarios. We list here some limitations of the model and avenues for further development that could be particularly interesting:

Depending on the context, additional interrelations in the Water-Energy-Food nexus, which are currently not simulated in the framework, can play an important role such as: energy consumption for water treatment or desalinisation (Dubreuil et al., 2013), energy for water pumping in the agricultural or domestic sector (Bauer-Gottwein et al., 2016; Dubreuil et al., 2013), water for cooling purposes of thermal power plants (Payet-Burin et al., 2018; Van Vliet et al., 2016) and production of crops for biofuels (Mirzabaev et al., 2015). For study cases where these interactions have an important impact, they can be added to the modelling framework.

In the next decades, renewables such as solar or wind energy will be crucial in the Southern African power sector (IRENA, 2013) and intermittence constraints will be a key element of future power systems. Hydropower plants have a lifetime of above 50 years and will therefore evolve among these future conditions. Hence, valuation of hydropower projects using a fixed price (e.g. Tilmant et al., 2012) or the concept of "firm energy" (e.g. World Bank, 2010) might no longer be appropriate (Palmintier, 2013). In this study, by using the concept of "load segments" (sometimes called "time slices"), we made a step towards the representation of intermittent energy systems, but a more detailed representation (considering e.g. ramping constraints, minimum loads, sun or wind profiles) will be key to correctly value hydropower projects.

In this study, we only considered the water resource in terms of quantity, however water quality may have an important impact for water treatment, irrigation, fishing, and tourism. Different approaches could be considered: (Boehlert et al., 2015) combine a water management model with a water quality model considering chemicals and reactions and represent advection among river branches, while Martinsen et al. (2018) consider water quality classes, with associated treatment costs and quality requirements for the demands, in a hydroeconomic optimization framework.

We presented the economic values of the development plans and their sensitivity to different sets of parameters but did not perform a complete Cost-Benefit Analysis of the projects. Costs and benefits linked to impacts on ecosystems, fishing, flood control, tourism, sedimentation and navigation need to be considered separately to complete a full Cost-Benefit Analysis of the infrastructure projects. Besides, some studies claim that the evaluation of investment costs, including financing, construction and resettlement costs are systematically and significantly underestimated (Ansar et al., 2014; Awojobi and Jenkins, 2015), which adds to the uncertainty in the net present value of the infrastructure projects.

By evaluating the development plans in the 2010, 2030 and 2050 scenarios, we showed that the timing of the investments plays an important role in an evolving socio-economic context. Furthermore, not all projects which are part of the development





plans may be profitable. Therefore, an important analysis would be the selection of the optimal projects, as well as timing and sequencing of investments, considering gradual changes in the socio-economic and climatic context.

The optimization framework of the model assumes full cooperation among different political and sectorial entities (e.g. upstream farmers in Zimbabwe may forgo some water abstractions to benefit Mozambique's downstream hydropower

production). The practical implementation of such trade-offs might be possible by using compensating payments (Tilmant et al., 2009), another approach is to consider trade-offs between efficiency and equity by using a multi-objective optimization (Hu et al., 2016). However, as this may be institutionally and politically complicated, decision makers might be interested in knowing the impacts on the planned projects if one or several countries do not cooperate. This could be implemented in the current modelling framework by solving the management decisions using a local objective function from upstream to

downstream.

Finally, we use a perfect foresight approach which is common to sectorial planning models (e.g. Kahil et al. (2018), Khan et al. (2018)). This means that optimal management decisions will anticipate future conditions such as droughts by storing additional water or cultivating crops with lower water requirements, leading to overestimation of system performance. In reality, water planners and managers will not have perfect foresight, and will be limited by the availability and skill of existing

forecasting systems. The validation of the model against observed indicators, shows that the bias due to perfect foresight assumption is not excessive. Furthermore, part of the bias is cancelled by doing relative analysis (e.g. with and without infrastructure development, with and without climate change scenario). However, as droughts have important economic impacts (SADC et al., 2015), a more realistic way of modelling reservoir operations and agriculture decisions could improve the reliability of the results. One way to implement this in the current modelling framework is to use Model Predictive Control

and iteratively solve the optimal management decisions in each time step with a limited knowledge of the future (Sahu, 2016).

## 6. Conclusion

We presented a new open-source decision support tool for economic valuation of water infrastructure and policies in the water-energy-food-climate nexus. The tool fills a gap in the existing planning tools, that are mostly single-resource focused, or do not have an optimization framework. Based on a hydroeconomic optimization framework, the tool considers synergies and

trade-offs among WEF infrastructure and policies and can be used to evaluate different scenarios.

In the Zambezi river basin, we show how the integrated analysis of the energy, agriculture, and water systems, including commodity markets, provides additional insights to the economic impacts of infrastructure and policies. This may lead to different investment decisions than those based on models not considering the nexus or market effects. We show that in a rapidly evolving socio-economic context and under potential pressure from climate change it is crucial to consider risks linked

to these uncertainties. In the driest climate change scenario, decrease in runoff reduces the hydropower production by 50%, causing losses of 700 million dollars per year, while rainfed agriculture is severely impacted by increased evapotranspiration and reduced rainfall, causing losses of about 1.6 billion dollars per year. The benefits of the hydropower development plan are





found to be around 1.9 billion dollars per year but are sensitive to future fuel prices or carbon pricing policies, capital costs of solar technologies and climate change. Similarly, the benefits of the irrigation development plan are found sensitive to the evolution of crop yields, world crop market prices and climate change. The development of irrigation infrastructure will decrease hydropower production, leading to reduced benefits. As the total water consumption is a limited share of the available

water, trade-offs represent only 5% of the value of the development plan. However, this effect could be exacerbated by climate change. Restoring natural flooding in the Zambezi delta involves limited economic trade-offs in the current climate, however under climate change it could result in major trade-offs with irrigation and hydropower generation.

*Code and data availability*

The decision support tool is available under the GNU General Public License version 3 (GPLv3) and can be downloaded with

the input data for the Zambezi study case from Github (https://github.com/RaphaelPB/WHAT-IF). The study case data are also available in (ReferenceXX), with the detailed sources.

*Appendices*

**A Linearization of the yield water response function**

The water requirement for a specific growth phase (ps) is estimated using the FAO 56 method (Allen et al., 1998), with the

reference evapotranspiration ($e_{T0}$) and a culture and phase specific crop coefficient ($k_c$). Therefore, considering the precipitation ($p$) and the amount of irrigation ($I_{rrig}$) during the growth phase, the crop demand satisfaction rate ($D_{rate}$) can be expressed as follow:

$$D_{rate}[ps] = \min(1, \frac{p[ps] + I_{rrig}[ps]}{k_c[ps] \cdot e_{T0}[ps]})$$

The relation between water demand satisfaction of cultures and yield is estimated using the additive yield water response

function based on the FAO 33 method (Doorenbos and Kassam, 1979). Crop production ($P_C$) is proportional to the product of the cultivated area ($A$) and maximum yield ($y$), corrected by the yield response factor ($k_Y$), which characterizes how the yield responds to water stress in the different growth phases (ps):

$$P_C = A \cdot y \cdot \left( 1 - \sum_{ps} k_Y[ps] \cdot (1 - D_{rate}[ps] ) \right)$$

For irrigated crops, the cultivated area ($A$) and the demand satisfaction rate ($D_{rate}$) are decision variables and therefore, the

equation is not linear as it is the product of the two. Considering four growth phases as defined by FAO (initial, development, medium and late), the number of possible combinations between the minimum and optimal demand satisfaction rates through the whole crop growth period is $2^4 = 16$. Consider now $m$ a 16x4 matrix of all combinations of minimum (0) and optimal (1) demand satisfactions per phase:





$$m = \begin{bmatrix} 1 & 1 & 1 & 1 \\ 0 & 1 & 1 & 1 \\ & & & \\ 0 & 0 & 0 & 0 \end{bmatrix}$$

We can now specify the crop production variable $P_C$ in an equation, linking the crop water demand satisfaction and the cultivated area in a single decision variable $A[pt]$, using the somewhat artificial notion that the farmer partitions his cultivated area into a selection of the 16 evapotranspiration combinations described by the path index pt. The overall demand satisfaction rate for each growth phase is the weighted average of the selected paths. Then the previous equation can be expressed as:

$$P_C = y \cdot \sum_{pt} \left( A[pt] \cdot \left(1 - \sum_{ps} k_Y[ps] \cdot (1 - m[pt, ps])\right) \right)$$

Which is a linear equation. Finally, for irrigated crops, the amount of irrigation ($I_{rrig}$) during a specific growth phase (ps) can be expressed as:

$$I_{rrig} = \sum_{pt} A[pt] \cdot \max(0, k_c \cdot e_{T0} \cdot m[pt, ps] - p)$$

Where $k_c \cdot e_{T0}$ and $p$ are respectively the crop water demand and precipitation during the growth phase.

## B Elastic demands

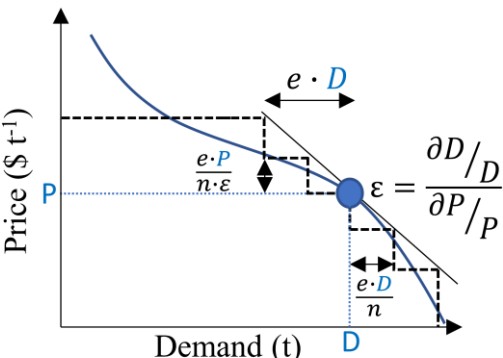

Observed point on demand curve
Real demand curve
Stepwise demand curve

**Figure B1: Stepwise representation of demand elasticity.** $\varepsilon$ represents elasticity, P, D are respectively the price and demand of the observed demand point, e and n are parameters of the stepwise function, e is the share of the demand that is elastic, and n is the number of steps. In the figure $\varepsilon = -1$, $n = 2$ and $e = 0.3$.



In order to represent the demand curve for crops, a demand point should be defined from observed data (e. g. FAO (2018)). If a demand elasticity is defined, the model will generate a stepwise demand curve representing the elasticity as shown in **Figure 1.** The stepwise function can be parametrized by setting $e$, the share of the demand that will be elastic and $nS$ the number of steps. Therefore, the Crop demand ($D_C$) and crop marginal value ($v_C$) parameters are divided into $1 + 2 \cdot nS$ steps as represented on the figure. Increasing the number of steps gives a finer approximation of the demand curve, however it increases the computation time as it increases the number of decision variables.

*Author contribution*

SPC, MK, KS and PBG designed the study, MK and RPB developed the computer model, RPB performed the analysis and wrote the manuscript, all authors contributed to the interpretation of results and commented on the manuscript.

*Competing Interests*

The authors declare that they have no conflict of interest.

*Acknowledgments*

Innovation Fund Denmark, COWIfonden, and COWI A/S founded the Industrial PhD project in which this research was carried out. The authors also thank Charles Fant for providing the data from the rainfall-runoff model.

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
