# Peer review of "WHAT-IF: an open-source decision support tool for water infrastructure investment planning within the Water-Energy-Food-Climate Nexus"

_Hydrology and Earth System Sciences, 2019_

## Referee Comment (RC1) · Anonymous Referee #1 · 1 Aug 2019

With great interest I have read this paper dealing with burning 'decision support' related questions. Not being a specialist in this specific field, it is difficult for me to properly assess the real added value of the manuscript. I can say that it seems to be quite an impressive work, the manuscript is well organised (maybe the case study presentation and results could be put together), it reads well and is very well written. Title and abstract reflect well the content of the paper, references seem appropriate. After a very clear introduction including a nice overview of the state-of-the-art, authors present their general modelling framework as well as an application including user-based scenarios

(please correct 'ReferenceXX' in the code availability section). The manuscript maybe rather long and I am wondering if the description of each modules (section 2) could be a supplementary material (?) in particular for Tables and Equations. As it is it may reads more like a report than a research article.

My main concern is the following; while there is a section discussing different scenarios I am missing some discussions on the sensitivity of each module to one of the other. Also the conclusion must be enhanced to reflect the large amount of work presented.

---

## Referee Comment (RC2) · Anonymous Referee #2 · 5 Aug 2019

In this article, an integrated water resource system model is developed to represent Water-Energy-Food-Climate Nexus. The paper is well written. I recommend publication of this paper with moderate revisions. My comments are as below: 1) It is not clear if this is an integrated water resource system model or a decision-support-tool. For instance, as the authors also mentioned decision-support-tools should provide a discussion platform to be used by different stakeholders. It is not clear how the developed model in this manuscript can achieve this goal. How user-friendly is this tool? Does it have a Graphical User Interface? 2) The literature on decision-support-tools should

be enriched. For example, see McIntosh, B. S., Ascough II, J. C., Twery, M., Chew, J., Elmahdi, A., Haase, D., ... & Chen, S. (2011). Environmental decision support systems (EDSS) development–challenges and best practices. Environmental Modelling & Software, 26(12), 1389-1402. 3) The novel contribution of this paper is not clear. 4) The authors discuss that the model captures Water-Energy-Food-Climate Nexus (shown in Figure 1). However, it is not clear how the developed model captures dynamic relationships among these elements. I suggest authors show graphically feedback loops within individual and among system elements. This can help understanding the model structure. 5) The nice part of this work is that the model is open source. However, the information on this feature needs more elaboration. How can users apply this model? What are the steps? What is the list of inputs to the model? 6) The model addresses the questions of "what-if" and "what is the best?" as it is an optimization model. Then why only "what-if" is used in the title? 7) The agricultural model needs more explanation and just referring to FAO methods is not enough. Is there any soil-moisture model? 8) The paper is really long and should be shortened.

---

## Author Comment (AC1) · 12 Aug 2019

We thank the reviewer for taking the time to review the manuscript and the pertinent comments. We have improved the manuscript following most suggestions and hope to have adequately answer your concerns. (please find also attached a pdf version of this reply with a color code)

**REVIEWER COMMENT 1** the manuscript is well organised (maybe the case study presentation and results could be put together)

[Figure]

AUTHOR REPPLY The arguments for keeping these sections apart are: 1) Avoid confusion between what is data input and what is results output 2) It provides to the reader with a work-flow example of how the model is used: 1 Assemble data, 2: Process results

**REVIEWER COMMENT 2** (please correct 'ReferenceXX' in the code availability section)

ACTION TAKEN We published the dataset as the zenodo archive https://zenodo.org/record/2646476#.XUmJ_XtS9O8 and added the correct reference (page 43, line 10): "The study case data are also available in (Payet-Burin, 2019), with the detailed sources." Payet-Burin, R.: Zambezi dataset to "WHAT-IF: an open-source decision support tool for water infrastructure investment planning within the Water-Energy-Food-Climate Nexus," zenodo.org, doi:10.5281/zenodo.2646476, 2019.

**REVIEWER COMMENT 3** The manuscript may be rather long and I am wondering if the description of each modules (section 2) could be a supplementary material (?) in particular for Tables and Equations. As it is it may reads more like a report than a research article.

ACTION TAKEN We agree with the suggestion to move the equations and parameters of each submodule of section 2 to a supplementary material document.

**REVIEWER COMMENT 4** My main concern is the following; while there is a section discussing different scenarios I am missing some discussions on the sensitivity of each module to one of the other.

AUTHOR REPPLY We understand this comment to be similar to comment 4) of reviewer 2, in the sense that the links between the different modules are not clear enough.

ACTION TAKEN To this purpose we added a figure showing the feedback loops among the modules, as suggested by reviewer 2 (page 17, line 9): "The main link in the nexus,
is the water resource for which hydropower, irrigation and ecosystems compete (Figure 2). The energy markets provide a dynamic value to hydropower production, while the crop markets provide a dynamic value of irrigation. The markets are therefore indirectly linked through the water trade-offs between hydropower and irrigation. Exogenous drivers on these markets such as new policies, technological and socio-economic changes, indirectly affect the water trade-offs and therefore all markets."

AUTHOR REPPLY – part 2 Regarding the quantitative sensitivity analysis on the Zambezi study case, the effect of the different modules on each-other are underlined at various points: The effect of climate change (water module) on the energy and agricultural system is discussed in the entire Section 4.2 Potential impacts of climate change p31-32. The effects of the Crop market module and the Energy module on the Agriculture production (page 37, line 7): "A drier climate has a twofold impact on the IDP (Figure 11): it reduces rainfed production and thus increases the value of irrigation, but it also increases trade-offs with the energy sector. In fact, in the current climate scenario the IDP saves 48 M\$ yr-1 of import value from the world crop market to satisfy food security constraints, while in the driest scenario it saves 95 M\$ yr-1 of import value. This shows the importance of representing rainfed agriculture to assess the value of irrigation projects. However, hydropower shortages induced by additional water consumption range from 515 GWh yr-1 in the wettest scenario to 1 600 GWh yr-1 in the driest scenario, inducing losses in the range of 24 to 104 M\$ yr-1 (representing up to more than 10% of the benefits) which counterbalance the import substitution effect in the crop market." The effects of Hydropower development on agricultural production (Page 33, line 22): "The HDP has no impact on the agricultural system (Table 15), neither positive or negative, and vice versa, the development of the irrigation development plan does almost not affect its value (Figure 9)." The effects of the environmental flows on the energy and agriculture systems (Page 40, line 14)): "Opportunity costs of the "base" environmental flow policy are almost zero except for the driest climate change scenario. The restoration of the natural floods induces increasing costs with the flood level target: costs reach up to more than 800 M\$ yr-1 for the driest scenario and the

highest flood level, but stay under 150 M$ yr-1 for the semi wet and wettest scenarios." As the reviewer points out this is not exactly a sensitivity analysis, however it shows the impacts of the respective modules on each-other. The sensitivity analysis is performed on the holistic solution for the exogenous parameters that are uncertain in the future such as energy and food demands, technologies capital costs, yields, climate change, carbon pricing or e-flow policies, crop world market prices. (Figure 9, Figure 11, Figure 12)

**REVIEWER COMMENT 5** Also the conclusion must be enhanced to reflect the large amount of work presented

ACTION TAKEN We agree that the conclusion could be extended and added the following (page 43, line 2): "The benefits of the hydropower development plan are found to be around 1.9 billion dollars per year but are sensitive to future fuel prices or carbon pricing policies, capital costs of solar technologies and climate change. Climate change is the main factor impacting hydropower production as it affects the water resource availability. A carbon pricing policy could have a significant impact on fuel prices and thus power production costs and is therefore the main driver on hydropower production value. The development of solar capacity will increase the intermittency in the power system and thus the value of hydropower, however it will decrease the cost of power production, and thus potentially counterbalance the first effect. Similarly, the benefits of the irrigation development plan are found sensitive to the evolution of crop yields, world crop market prices and climate change. The potential improvements in yields could have significant positive impact on the crop production, however the increase is uncertain as past data does not show a clear improving trend. As most of the value of the irrigation development is generated through exports, the development plan is very sensitive to world crop market prices. A dryer climate will reduce the availability of water and thus the potential benefits, however it also increases the value of crops during dry years as rainfed crops will be affected. The development of irrigation infrastructure will decrease hydropower production, leading to reduced

benefits. As the total water consumption is a limited share of the available water, trade-offs represent only 5% of the value of the development plan. However, this effect could be exacerbated by climate change."

Please also note the supplement to this comment:
https://www.hydrol-earth-syst-sci-discuss.net/hess-2019-167/hess-2019-167-AC1-supplement.pdf

[Figure]

[Figure]

**Fig. 1.**

---

## Author Comment (AC2) · 12 Aug 2019

We thank the reviewer for taking the time to review the manuscript and the pertinent comments. We have improved the manuscript following most suggestions and clarified some interrogations. (The reply is also available as a pdf supplement including a color code)

\*\*REVIEWER COMMENT 1\*\* 1) It is not clear if this is an integrated water resource system model or a decision-support-tool. For instance, as the authors also mentioned

decision-support-tools should provide a discussion platform to be used by different stakeholders. It is not clear how the developed model in this manuscript can achieve this goal. How user-friendly is this tool? Does it have a Graphical User Interface?

AUTHOR REPLY The reviewer is mentioning an important feature of a decision support-tool: how can it be used/implemented in practice, using stakeholder participation. While WHAT-IF is intended to be a decision support tool, this paper is the description of the scientific base of the integrated water resource system model. However, the Zambezi study case shows how the model is able to answer typical questions that will support decision making (Section 4, page 28, line 5): "In this section, we illustrate how the Zambezi model can be used to answer questions such as "What are the potential impacts of climate change on the agriculture and energy systems?", "What are the benefits of the hydropower and agricultural development plans?", "What is the sensitivity of these benefits regarding uncertainties in policies, future climate and socio-economic development ?", "What are the synergies and trade-offs between the irrigation and hydropower development plan?", and "What are the opportunity costs of restoring flood regimes in the Zambezi delta ?" " By being a community based open-source framework, the idea is that further features will follow on the GitHub repository, but are not part of this publication,

ACTION TAKEN we clarify this and add the suggested reference in comment 2) (page 6, line2): "For this reason, the model is holistic in its resolution, but modular in its formulation, the user can activate or deactivate different modules and new modules representing relevant interrelations are easy to add. The flexibility of the framework and the open-source character will enable the tool to evolve with user and stakeholder inputs. Additional features will be added such as GIS visualization and data acquisition modules; Mcintosh et al., (2011) describes some of the challenges and best practices of developing an environmental decision support system."

AUTHOR REPLY – part 2 The current graphical interface is excel spreadsheets

ACTION TAKEN - part 2 we clarify this (page 5, line 7): The model can be connected to different open-source or commercial solvers; input data and output results are organized in MS Excel spreadsheets.

**REVIEWER COMMENT 2** 2) The literature on decision-support-tools should be enriched. For example, see McIntosh, B. S., Ascough II, J. C., Twery, M., Chew, J.,Elmahdi, A., Haase, D., ... & Chen, S. (2011). Environmental decision support systems (EDSS) development–challenges and best practices. Environmental Modelling & Soft-ware, 26(12), 1389-1402.

ACTION TAKEN We implemented this in the answer of comment 1)

**REVIEWER COMMENT 3** 3) The novel contribution of this paper is not clear.

AUTHOR REPPLY The novel contributions pointed out in the article is the combination of these 3 elements: 1-The representation of the agricultural and power markets in a hydro-economic model (page 2, line 28): "Traditionally, agricultural and energy water users are represented with an exogenous demand and willingness-to-pay for water (Bauer-Gottwein et al., 2017). Therefore, classic hydroeconomic models are able to analyse trade-offs and synergies between water users, but are not as effective in terms of representing dynamic interactions between infrastructure, policies, and commodity markets." 2-The spatial and temporal scale of the water representation in a nexus model (page 2, line 32): "On the other hand, nexus models, particularly energy centred models (e.g. OSeMOSYS (Howells et al., 2011) and TIAM-FR (Dubreuil et al., 2013)) tend to ignore the spatial and temporal scale of water availability and therefore may overlook water scarcity problems (Khan et al., 2017)." 3-The optimization framework (page 4, line 3): "In contrast to simulation models that are rule-based (such as WEAP), the model finds the optimal water, agriculture and energy management decisions, considering trade-offs and synergies between them." The last novelty is the application of this framework to the Zambezi River Basin: Section 3, (page 17, line 15) to (page 18, line 15) describes how this study is different from the other similar studies

in the Zambezi river basin.

ACTION TAKEN As the reviewer points out, this is not explicit enough, we suggest making it more explicit (page 3, line 8): In this study, we developed a new open-source decision support tool for water infrastructure investment planning. The novelty of the tool is that it combines a hydro-economic optimization framework, with a nexus representation of the agriculture and food systems. The tool can represent political boundaries, the joint development of WEF infrastructure and policies, and uncertainty in future climate and socio-technical changes.

**REVIEWER COMMENT 4** 4) The authors discuss that the model captures Water-Energy-Food-Climate Nexus (shown in Figure 1). However, it is not clear how the developed model captures dynamic relationships among these elements. I suggest authors show graphically feedback loops within individual and among system elements. This can help understanding the model structure.

ACTION TAKEN We agree with the reviewer that a figure would clarify the nexus interactions, and add the following figure and text (page 17, line 9): "The main link in the nexus, is the water resource for which hydropower, irrigation and ecosystems compete (Figure 2). The energy markets provide a dynamic value to hydropower production, while the crop markets provide a dynamic value of irrigation. The markets are therefore indirectly linked through the water trade-offs between hydropower and irrigation. Exogenous drivers on these markets such as new policies, technological and socio-economic changes, indirectly affect the water trade-offs and therefore all markets."

Figure 2: Main feedback loops in the water-energy-food nexus representation. All flows are holistically solved to maximize total economic surplus, the water, energy and crop values are the resulting duals of the mass balances constraints. The figure does not show the temporal and spatial scale of the nexus problem.

**REVIEWER COMMENT 5** 5) The nice part of this work is that the model is open source. However, the information on this feature needs more elaboration. How can

users apply this model? What are the steps? What is the list of inputs to the model?

AUTHOR REPLY The practical use of the model is intended to be described in the github repository, the link is provided in Code and data availability page 43. For the list of inputs the reader can refer to the table of parameters within the equations (Tables 1 to 5), and the Section 3: Zambezi river basin study case, shows the data collection for a specific case.

ACTION TAKEN We add a missing reference to the github repository (page 5, line 7): The code and installation instructions can be found on Github (https://github.com/RaphaelPB/WHAT-IF) A document named "INSTALLING_WHATIF" guiding through the steps to install the tool has been added in the "Documents" folder of the github repository.

**REVIEWER COMMENT 6** 6) The model addresses the questions of "what-if" and "what is the best?" as it is an optimization model. Then why only "what-if" is used in the title?

AUTHOR REPLY As mentioned the optimization framework is mainly a way of simulating a resource management that adapts to changing conditions (page 4, line 4): "The optimization framework simulates adaptation to new infrastructure and policies, climate change, and socio-economic development. Conversely, in a rule-based simulation framework, allocation rules are usually based on the current socio-economic conditions or new rules are estimated, which may lead to suboptimal allocation decisions and underestimation of project benefits (Pereira-Cardenal et al., 2016)." A part of this, WHAT-IF stands for Water, Hydropower, Agriculture Tool for Investment and Financing, but this is only mentioned in the Github repository

ACTION TAKEN We therefore add the acronym signification within the article (page 3, line 9): In this study, we developed a new open-source decision support tool for water infrastructure investment planning, based on a hydroeconomic optimization model in a nexus framework: WHAT-IF, Water, Hydropower, Agriculture Tool for Investment and

Financing.

**REVIEWER COMMENT 7** 7) The agricultural model needs more explanation and just referring to FAO methods is not enough. Is there any soil-moisture model?

AUTHOR REPLY Section 2.2 Agriculture production, page 9 to 11; details all equations, variables and parameters used in the representation of the agriculture system (with a reference to the appendices for the yield water response function). Soil moisture is not accounted for in the FAO 56 formula that we use. The assumption of the formula is that it has little impact at the monthly/growing season time-scale, the IMPACT model (by IFPRI) did the same assumption in its 2008 version. We might consider it for further version, otherwise a way around is to add it in the form of "net precipitation".

**REVIEWER COMMENT 8** 8) The paper is really long and should be shortened

ACTION TAKEN As suggested as well by reviewer 1, we move the equations and parameters of each submodule of section 2 to a supplementary material document.

Please also note the supplement to this comment:
https://www.hydrol-earth-syst-sci-discuss.net/hess-2019-167/hess-2019-167-AC2-supplement.pdf

———————————————

[Figure]

**Fig. 1.**

---

## Author Response (AR1)

**REPLY TO REVIEWERS**

We thank the reviewer for taking the time to review the manuscript and the pertinent comments. We have improved the manuscript following most suggestions and clarified some interrogations. In the reply to the reviewer we use the following color code:

reviewer's comment authors' answer *extract from the article modification to the article* Action taken

**REVIEWER 1:**

the manuscript is well organised (maybe the case study presentation and results could be put together),

The arguments for keeping these sections apart are:

1) Avoid confusion between what is data input and what is results output

2) It provides to the reader with a work-flow example of how the model is used: 1 Assemble data, 2: Process results

(please correct 'ReferenceXX' in the code availability section)

We published the dataset as the zenodo archive https://zenodo.org/record/2646476#.XUmJ\_XtS9O8 and added the correct reference (page 43, line 10):

The study case data are also available in (Payet-Burin, 2019), with the detailed sources.

Payet-Burin, R.: Zambezi dataset to "WHAT-IF: an open-source decision support tool for water infrastructure investment planning within the Water-Energy-Food-Climate Nexus," zenodo.org, doi:10.5281/zenodo.2646476, 2019.

The manuscript may be rather long and I am wondering if the description of each modules (section 2) could be a supplementary material (?) in particular for Tables and Equations. As it is it may reads more like a report than a research article.

We agree with the suggestion to move the equations and parameters of each submodule of section 2 to a supplementary material document.

My main concern is the following; while there is a section discussing different scenarios I am missing some discussions on the sensitivity of each module to one of the other.

We understand this comment to be similar to comment 4) of reviewer 2, in the sense that the links between the different modules are not clear enough.

To this purpose we added a figure showing the feedback loops among the modules, as suggested by reviewer 2.

Regarding the quantitative sensitivity analysis on the Zambezi study case, the effect of the different modules on each-other are underlined at various points:

The effect of climate change (water module) on the energy and agricultural system is discussed in the entire Section 4.2 Potential impacts of climate change p31-32.

The effects of the Crop market module and the Energy module on the Agriculture production (page 37, line 7):

A drier climate has a twofold impact on the IDP (Figure 11): it reduces rainfed production and thus increases the value of irrigation, but it also increases trade-offs with the energy sector. In fact, in the current climate

scenario the IDP saves 48 M\$ yr1 of import value from the world crop market to satisfy food security constraints, while in the driest scenario it saves 95 M\$ yr1 of import value. This shows the importance of representing rainfed agriculture to assess the value of irrigation projects. However, hydropower shortages induced by additional water consumption range from 515 GWh yr1 in the wettest scenario to 1 600 GWh yr1 in the driest scenario, inducing losses in the range of 24 to 104 M\$ yr1 (representing up to more than 10% of the benefits) which counterbalance the import substitution effect in the crop market.

The effects of Hydropower development on agricultural production (Page 33, line 22):

The HDP has no impact on the agricultural system (Table 15), neither positive or negative, and vice versa, the development of the irrigation development plan does almost not affect its value (Figure 9).

The effects of the environmental flows on the energy and agriculture systems (Page 40, line 14)):

Opportunity costs of the "base" environmental flow policy are almost zero except for the driest climate change scenario. The restoration of the natural floods induces increasing costs with the flood level target: costs reach up to more than 800 M\$  $yr^1$  for the driest scenario and the highest flood level, but stay under 150 M\$  $yr^1$  for the semi wet and wettest scenarios.

As the reviewer points out this is not exactly a sensitivity analysis, however it shows the impacts of the respective modules on each-other. The sensitivity analysis is performed on the holistic solution for the exogenous parameters that are uncertain in the future such as energy and food demands, technologies capital costs, yields, climate change, carbon pricing or e-flow policies, crop world market prices. (Figure 9, Figure 11, Figure 12)

Also the conclusion must be enhanced to reflect the large amount of work presented

**We agree that the conclusion could be extended and added the following (page 43, line 2):**

The benefits of the hydropower development plan are found to be around 1.9 billion dollars per year but are sensitive to future fuel prices or carbon pricing policies, capital costs of solar technologies and climate change. Climate change is the main factor impacting hydropower production as it affects the water resource availability. A carbon pricing policy could have a significant impact on fuel prices and thus power production costs and is therefore the main driver on hydropower production value. The development of solar capacity will increase the intermittency in the power system and thus the value of hydropower, however it will decrease the cost of power production, and thus potentially counterbalance the first effect. Similarly, the benefits of the irrigation development plan are found sensitive to the evolution of crop yields, world crop market prices and climate change. The potential improvements in yields could have significant positive impact on the crop production, however the increase is uncertain as past data does not show a clear improving trend. As most of the value of the irrigation development is generated through exports, the development plan is very sensitive to world crop market prices. A dryer climate will reduce the availability of water and thus the potential benefits, however it also increases the value of crops during dry years as rainfed crops will be affected. The development of irrigation infrastructure will decrease hydropower production, leading to reduced benefits. As the total water consumption is a limited share of the available water, tradeoffs represent only 5% of the value of the development plan. However, this effect could be exacerbated by climate change.

**REVIEWER 2:**

1) It is not clear if this is an integrated water resource system model or a decision-support-tool. For instance, as the authors also mentioned decision-support-tools should provide a discussion platform to be used by different stakeholders. It is not clear how the developed model in this manuscript can achieve this goal. How user-friendly is this tool? Does it have a Graphical User Interface?

The reviewer is mentioning an important feature of a decision support-tool: how can it be used/implemented in practice, using stakeholder participation. While WHAT-IF is intended to be a decision support tool, this paper is the description of the scientific base of the integrated water resource system model. However, the Zambezi study case shows how the model is able to answer typical questions that will support decision making (Section 4, page 28, line 5):

In this section, we illustrate how the Zambezi model can be used to answer questions such as "What are the potential impacts of climate change on the agriculture and energy systems?", "What are the benefits of the hydropower and agricultural development plans?", "What is the sensitivity of these benefits regarding uncertainties in policies, future climate and socio-economic development ?", "What are the synergies and trade-offs between the irrigation and hydropower development plan?", and "What are the opportunity costs of restoring flood regimes in the Zambezi delta ?"

By being a community based open-source framework, the idea is that further features will follow on the GitHub repository, but are not part of this publication,

we clarify this and add the suggested reference in comment 2) (page 6, line2):

For this reason, the model is holistic in its resolution, but modular in its formulation, the user can activate or deactivate different modules and new modules representing relevant interrelations are easy to add. Mcintosh et al. (2011) describes some of the challenges and best practices of developing an environmental decision support system, it includes: start simple and small with a modular approach, plan for longevity with a framework easy to update, design for ease of use including a user-friendly interface, and design for usefulness by including stakeholders' input. Following these recommendations, the flexibility of the framework and its open-source character will enable the tool to evolve with user and stakeholder inputs and additional features will be added such as GIS visualization and data acquisition modules.

The current graphical interface is excel spreadsheets

we clarify this (page 5, line 7):

The model can be connected to different open-source or commercial solvers; input data and output results are organized in MS Excel spreadsheets.

2) The literature on decision-support-tools should be enriched. For example, see McIntosh, B. S., Ascough II, J. C., Twery, M., Chew, J., Elmahdi, A., Haase, D., ... & Chen, S. (2011). Environmental decision support systems (EDSS) development–challenges and best practices. Environmental Modelling & Soft-ware, 26(12), 1389-1402.

We implemented this in the answer of comment 1)

3) The novel contribution of this paper is not clear.

The novel contributions pointed out in the article is the combination of these 3 elements:

1-The representation of the agricultural and power markets in a hydro-economic model (page 2, line 28):

Traditionally, agricultural and energy water users are represented with an exogenous demand and willingness-to-pay for water (Bauer-Gottwein et al., 2017). Therefore, classic hydroeconomic models are able to analyse trade-offs and synergies between water users, but are not as effective in terms of representing dynamic interactions between infrastructure, policies, and commodity markets.

2-The spatial and temporal scale of the water representation in a nexus model (page 2, line 32):

On the other hand, nexus models, particularly energy centred models (e.g. OSeMOSYS (Howells et al., 2011) and TIAM-FR (Dubreuil et al., 2013)) tend to ignore the spatial and temporal scale of water availability and therefore may overlook water scarcity problems (Khan et al., 2017).

3-The optimization framework (page 4, line 3):

In contrast to simulation models that are rule-based (such as WEAP), the model finds the optimal water, agriculture and energy management decisions, considering trade-offs and synergies between them.

The last novelty is the application of this framework to the Zambezi River Basin: Section 3, (page 17, line 15) to (page 18, line 15) describes how this study is different from the other similar studies in the Zambezi river basin.

As the reviewer points out, this is not explicit enough, we suggest making it more explicit (page 3, line 8):

In this study, we developed a new open-source decision support tool for water infrastructure investment planning. The novelty of the tool is that it combines a hydro-economic optimization framework, with a nexus representation of the agriculture and food systems. The tool can represent political boundaries, the joint development of WEF infrastructure and policies, and uncertainty in future climate and socio-technical changes.

4) The authors discuss that the model captures Water-Energy-Food-Climate Nexus (shown in Figure 1). However, it is not clear how the developed model captures dynamic relationships among these elements. I suggest authors show graphically feedback loops within individual and among system elements. This can help understanding the model structure.

We agree with the reviewer that a figure would clarify the nexus interactions, and add the following figure and text (page 17, line 9):

The main link in the nexus, is the water resource for which hydropower, irrigation and ecosystems compete (Figure 2). The energy markets provide a dynamic value to hydropower production, while the crop markets provide a dynamic value of irrigation. The markets are therefore indirectly linked through the water trade-offs between hydropower and irrigation. Exogenous drivers on these markets such as new policies, technological and socio-economic changes, indirectly affect the water trade-offs and therefore all markets.

*Figure 2: Main feedback loops in the water-energy-food nexus representation.* All flows are holistically solved to maximize total economic surplus, the water, energy and crop values are the resulting duals of the mass balances constraints. The figure does not show the temporal and spatial scale of the nexus problem.

5) The nice part of this work is that the model is open source. However, the information on this feature needs more elaboration. How can users apply this model? What are the steps? What is the list of inputs to the model?

The practical use of the model is intended to be described in the github repository, the link is provided in *Code and data availability* page 43.

We add a missing reference to the github repository (page 5, line 7):

The code and installation instructions can be found on Github (https://github.com/RaphaelPB/WHAT-IF)

A document named "INSTALLING\_WHATIF" guiding through the steps to install the tool has been added in the "Documents" folder of the github repository.

For the list of inputs the reader can refer to the table of parameters within the equations (Tables 1 to 5), and the Section 3: Zambezi river basin study case, shows the data collection for a specific case.

6) The model addresses the questions of "what-if" and "what is the best?" as it is an optimization model. Then why only "what-if" is used in the title?

As mentioned the optimization framework is mainly a way of simulating a resource management that adapts to changing conditions (page 4, line 4):

The optimization framework simulates adaptation to new infrastructure and policies, climate change, and socio-economic development. Conversely, in a rule-based simulation framework, allocation rules are usually based on the current socio-economic conditions or new rules are estimated, which may lead to suboptimal allocation decisions and underestimation of project benefits (Pereira-Cardenal et al., 2016).

A part of this, WHAT-IF stands for Water, Hydropower, Agriculture Tool for Investment and Financing, but this is only mentioned in the Github repository

We therefore add the acronym signification within the article (page 3, line 9):

In this study, we developed a new open-source decision support tool for water infrastructure investment planning, based on a hydroeconomic optimization model in a nexus framework: WHAT-IF, Water, Hydropower, Agriculture Tool for Investment and Financing.

7) The agricultural model needs more explanation and just referring to FAO methods is not enough. Is there any soil-moisture model?

Section 2.2 Agriculture production, page 9 to 11; details all equations, variables and parameters used in the representation of the agriculture system (with a reference to the appendices for the yield water response function).

Soil moisture is not accounted for in the FAO 56 formula that we use. The assumption of the formula is that it has little impact at the monthly/growing season time-scale, the IMPACT model (by IFPRI) did the same assumption in its 2008 version. We might consider it for further version, otherwise a way around is to add it in the form of "net precipitation".

8) The paper is really long and should be shortened

As suggested as well by reviewer 1, we move the equations and parameters of each submodule of section 2 to a supplementary material document.

[revised manuscript text omitted]